# A Brownian ratchet model for DNA loop extrusion by the cohesin complex

**Torahiko L Higashi[1], Georgii Pobegalov[2,3], Minzhe Tang[1], Maxim I Molodtsov[2,3]\*, Frank Uhlmann[1]\***

[1]Chromosome Segregation Laboratory, The Francis Crick Institute, London, United Kingdom; [2]Mechanobiology and Biophysics Laboratory, The Francis Crick Institute, London, United Kingdom; [3]Department of Physics and Astronomy, University College London, London, United Kingdom

**Abstract** The cohesin complex topologically encircles DNA to promote sister chromatid cohesion. Alternatively, cohesin extrudes DNA loops, thought to reflect chromatin domain formation. Here, we propose a structure-based model explaining both activities. ATP and DNA binding promote cohesin conformational changes that guide DNA through a kleisin N-gate into a DNA gripping state. Two HEAT-repeat DNA binding modules, associated with cohesin's heads and hinge, are now juxtaposed. Gripping state disassembly, following ATP hydrolysis, triggers unidirectional hinge module movement, which completes topological DNA entry by directing DNA through the ATPase head gate. If head gate passage fails, hinge module motion creates a Brownian ratchet that, instead, drives loop extrusion. Molecular-mechanical simulations of gripping state formation and resolution cycles recapitulate experimentally observed DNA loop extrusion characteristics. Our model extends to asymmetric and symmetric loop extrusion, as well as z-loop formation. Loop extrusion by biased Brownian motion has important implications for chromosomal cohesin function.

**\*For correspondence:**
m.molodtsov@ucl.ac.uk (MIM);
frank.uhlmann@crick.ac.uk (FU)

**Competing interests:** The authors declare that no competing interests exist.

## Introduction

Cohesin is a member of the Structural Maintenance of Chromosomes (SMC) family of ring-shaped chromosomal protein complexes that are central to higher order chromosome organization (*Hirano, 2016*; *Uhlmann, 2016*; *Yatskevich et al., 2019*). Cohesin holds together replicated sister chromatids from the time of their synthesis in S phase, until mitosis, to ensure their faithful segregation during cell divisions (*Guacci et al., 1997*; *Michaelis et al., 1997*). In addition, budding yeast cohesin participates in mitotic chromosome condensation, while higher eukaryotic cohesin impacts gene regulation by defining boundary elements during interphase chromatin domain formation (*Parelho et al., 2008*; *Wendt et al., 2008*). Cohesin is also recruited to sites of double-stranded DNA breaks to promote DNA repair by homologous recombination and to stalled DNA replication forks to aid restart of DNA synthesis (*Ström et al., 2004*; *Unal et al., 2004*; *Tittel-Elmer et al., 2012*). Understanding how cohesin carries out all these biological functions requires the elucidation of the molecular mechanisms by which cohesin interacts with DNA, as well as how cohesin establishes interactions between more than one DNA.

Cohesin's DNA binding activity is contained within its unique ring architecture (*Gligoris et al., 2014*; *Huis in 't Veld et al., 2014*). Two SMC subunits, Smc1[Psm1] and Smc3[Psm3], form long flexible coiled coils that are connected at one end by a dimerization interface known as the hinge (generic gene names are accompanied by fission yeast subunit names in superscript; fission yeast cohesin was used for the experiments and structural analyses in this study, see *Figure 1A*). At the other end lie ABC transporter-type ATPase head domains whose dimerization is regulated by ATP binding. The two SMC heads are further connected by a kleisin subunit, Scc1[Rad21]. The kleisin N-terminus

**eLife digest** When a cell divides, it has to ensure that each of its daughter cells inherits one copy of its genetic information. It does this by duplicating its chromosomes (the DNA molecules that encode the genome) and distributing one copy of each to its daughter cells. Once a cell duplicates a chromosome, the two identical chromosomes must be held together until the cell is ready to divide in two. A ring-shaped protein complex called cohesin does this by encircling the two chromosomes. Cohesin embraces both chromosome copies, as they emerge from the DNA replicating machinery. The complex is formed of several proteins that bind to a small molecule called ATP, whose arrival and subsequent breakdown release energy.

Cohesin also interacts with DNA in a different way: it can create loops of chromatin (the complex formed by DNA and its packaging proteins) that help regulate the activity of genes. Experiments performed on single molecules isolated in the laboratory show that cohesin can form a small loop of DNA that is then enlarged through a process called DNA loop extrusion. However, it is not known whether loop extrusion occurs in the cell.

Although both of cohesin's roles have to do with how DNA is organised in the cell, it remains unclear how a single protein complex can engage in two such different activities. To answer this question, Higashi et al. used a structure of cohesin from yeast cells gripping onto DNA to build a model that simulates how the complex interacts with chromosomes and chromatin. This model suggested that when ATP is broken down, the cohesin structure shifts and DNA enters the ring, allowing DNA to be entrapped and chromosomes to be bound together. However, a small change in how DNA is gripped initially could prevent it from entering the ring, creating a ratchet mechanism that forms and enlarges a DNA loop.

This molecular model helps explain how cohesin can either encircle DNA or create loops. However, Higashi et al.'s findings also raise the question of whether loop extrusion is possible inside cells, where DNA is densely packed and bound to proteins which could be obstacles to loop extrusion. Further research to engineer cohesin that can only perform one of these roles would help to clarify their individual contributions in the cell.

---

reversibly engages with Smc3$^{Psm3}$ coiled coil next to the ATPase head, forming the kleisin N-gate through which DNA enters the cohesin ring (*Figure 1A*; *Higashi et al., 2020*). The kleisin C-terminus in turn binds to the Smc1 head domain. These kleisin terminal domains are connected via a long unstructured region, to which two HEAT repeat subunits bind that promote topological cohesin loading onto DNA. Scc3$^{Psc3}$ interacts with the middle of the kleisin unstructured region. Scc2$^{Mis4}$, together with its binding partner Scc4$^{Ssl3}$, transiently associates with the kleisin between the kleisin N-gate and Scc3$^{Psc3}$. Once cohesin loading onto DNA is complete, Scc2$^{Mis4}$ is replaced by a related HEAT repeat subunit, Pds5 (*Murayama and Uhlmann, 2015*; *Petela et al., 2018*). Because of its transient role, Scc2$^{Mis4}$-Scc4$^{Ssl3}$ is often thought of as a cofactor, termed 'cohesin loader' (*Ciosk et al., 2000*; *Murayama and Uhlmann, 2014*). Following topological loading, cohesin is free to linearly diffuse along DNA in vitro (*Davidson et al., 2016*; *Kanke et al., 2016*; *Stigler et al., 2016*), while RNA polymerases push cohesin along chromosomes toward sites of transcriptional termination in vivo (*Lengronne et al., 2004*; *Ocampo-Hafalla et al., 2016*; *Busslinger et al., 2017*). Cohesin promotes sister chromatid cohesion following DNA replication by topologically entrapping two sister DNAs (*Haering et al., 2008*; *Murayama et al., 2018*).

In addition to topologically entrapping DNA, in vitro experiments have revealed the ability of human cohesin to translocate along DNA in a directed motion, as well as its ability to extrude DNA loops (*Davidson et al., 2019*; *Kim et al., 2019*). These activities are reminiscent of those previously observed with a related SMC complex, condensin, a central mediator of mitotic chromosome condensation (*Terakawa et al., 2017*; *Ganji et al., 2018*). Like topological loading onto DNA, loop extrusion by cohesin depends on its ATPase, as well as on the human Scc3$^{Psc3}$ homolog SA1 and the cohesin loader (NIPBL-MAU2). In contrast to topological loading, cohesin is able to extrude DNA loops if all three cohesin ring interfaces are covalently closed (*Davidson et al., 2019*). This suggests that loop extrusion does not involve topological DNA entry into the cohesin ring.

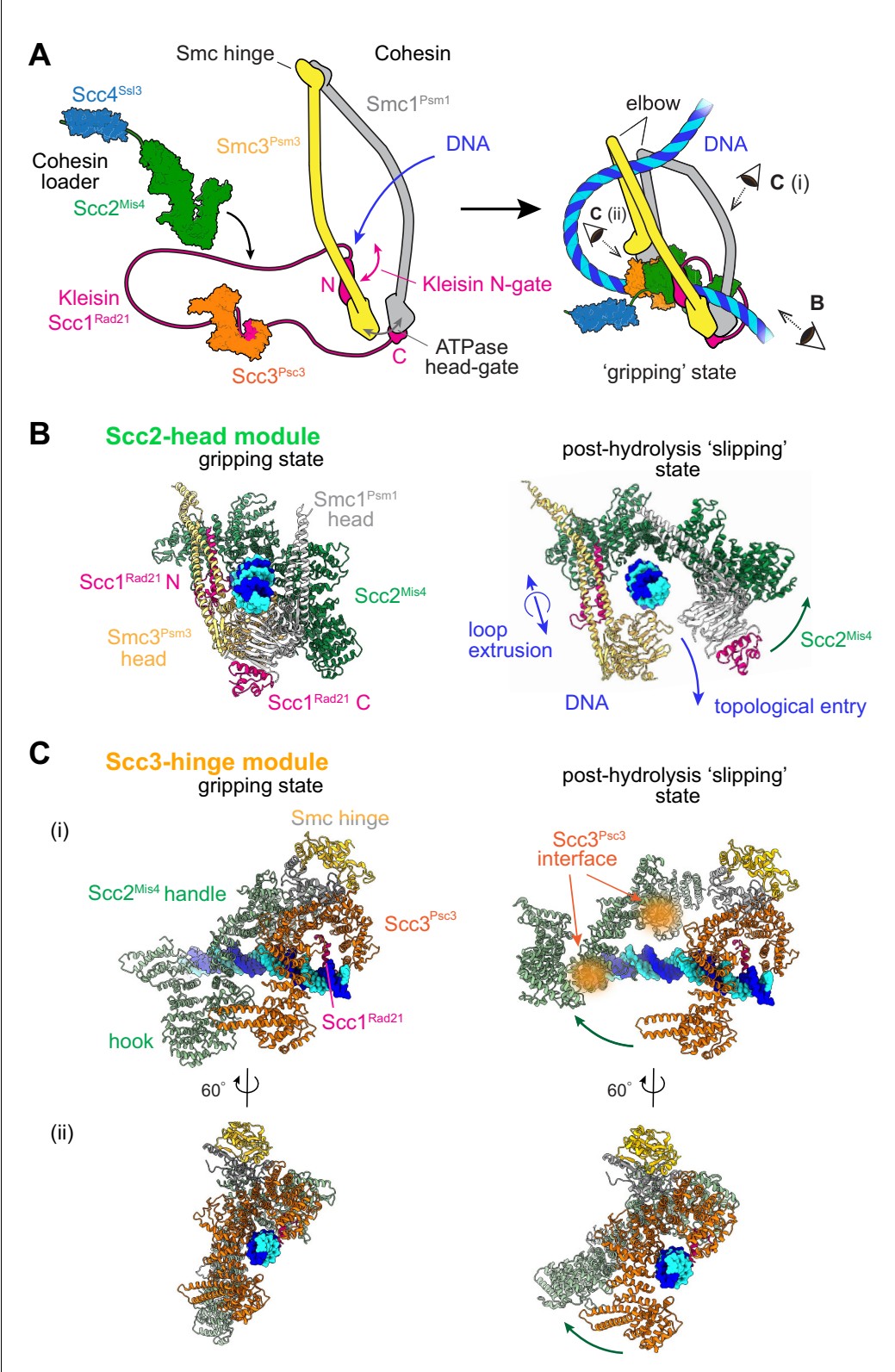

**Figure 1.** Two cohesin DNA binding modules in the gripping state and in their projected post-hydrolysis states. (**A**) Structural overview of the cohesin complex components and assembly of the DNA gripping state. Viewpoints of the structures shown in (**B** and **C**) are indicated. (**B**) The Scc2-head module in the gripping state and its predicted conformation following ATP hydrolysis. ATPase head disengagement and return of Scc2[Mis4] from the gripping state conformation to its extended crystal structure form results in loss of DNA interactions, turning the module into its 'slipping state'. (**C**) The

*Figure 1 continued on next page*

*Figure 1 continued*

Scc3-hinge module. The DNA binding surface of this module remains unaltered by ATP hydrolysis. However, the Scc2^Mis4 conformational change leads to uncoupling of the Scc3-hinge from the Scc2-head module.

The online version of this article includes the following figure supplement(s) for figure 1:

**Figure supplement 1.** Configuration of Scc3^Psc3 and Scc2^Mis4 in the DNA gripping state.

Several models have been proposed as to how SMC complexes extrude DNA loops. These include a tethered-inchworm model in which a scissoring motion of the ATPase heads translates into movement along DNA (*Nichols and Corces, 2018*). The DNA-segment-capture model instead suggests that a pumping motion between open and closed configurations of the SMC coiled coils constrains DNA loops (*Marko et al., 2019*). Finally, a scrunching model proposes that the SMC hinge reaches out to capture and reel in DNA loops (*Ryu et al., 2020a*). A characteristic of experimentally observed loop extrusion is that very small counterforces (<1 pN) stall loop growth (*Ganji et al., 2018*; *Golfier et al., 2020*). Both the DNA-segment-capture and the scrunching models therefore suggest that diffusive DNA motion contributes to loop growth. However, molecular details how SMC complexes enact these loop extrusion mechanisms remain elusive. Another important open question remains whether SMC complexes can move along and extrude physiological chromatin substrates, densely decorated by histones and other DNA binding proteins and folded into higher order structures (*Nozaki et al., 2017*; *Xu et al., 2018*).

We recently solved a cryo-EM structure of fission yeast cohesin with its loader in a nucleotide-bound DNA gripping state (*Higashi et al., 2020*). Together with DNA-protein crosslinking and biophysical experiments, this allowed us to trace the DNA trajectory into the cohesin ring by sequential passage through a kleisin N-gate and an ATPase head gate. We noticed, however, that kleisin N-gate passage might not be a strict prerequisite for DNA to reach the gripping state. We speculated that an alternative gripping state arises in which DNA has not passed the kleisin N-gate. While topological DNA entry is barred, this state might constitute an intermediate during loop extrusion. Here, we show how two DNA binding modules of the cohesin complex, formed by its HEAT repeat subunits (*Murayama and Uhlmann, 2014*; *Li et al., 2018*; *Collier et al., 2020*; *Kurokawa and Murayama, 2020*), are juxtaposed in the gripping state but swing apart following ATP hydrolysis. We illustrate how this swinging motion promotes completion of topological DNA entry, or alternatively generates DNA movements. Computational simulations of the latter scenario demonstrate how a Brownian ratchet forms that can drive loop extrusion. Our study provides a molecular proposal for both topological entry into the cohesin ring as well as for DNA loop extrusion.

## Results

### Two DNA binding modules in the cohesin-DNA gripping state

Two cryo-EM structures of fission yeast and human cohesin in the presence of non-hydrolyzable ATP analogs *Higashi et al., 2020*; *Shi et al., 2020* have revealed how the cohesin complex components come together during topological loading onto DNA (*Figure 1A*). ATP binding to the ATPase heads opens up a kleisin N-gate, allowing DNA to enter and reach the top of the engaged ATPase heads. The Scc2^Mis4 cohesin loader subunit then embraces the DNA, thereby closing the kleisin N-gate again and forming the DNA 'gripping state' that is captured by the cryo-EM structures (*Figure 1B*, left). The DNA is held in place by numerous positively charged surface residues, contributed by both the SMC heads and Scc2^Mis4. The cohesin loader has undergone a marked conformational change, compared to its crystallographically observed 'extended' form. Its N-terminal handle embraces the DNA in the gripping state, thereby adopting a 'bent' conformation. We refer to the composite DNA interaction site, consisting of Scc2^Mis4 and the SMC ATPase heads, as the 'Scc2-head module'.

A second DNA contact in the gripping state, situated behind the cohesin loader, is made by Scc3^Psc3 in conjunction with the kleisin middle region (*Figure 1A* and *Figure 1C*, left). The SMC hinge touches down to bridge cohesin loader and Scc3^Psc3, enabled by SMC coiled coil inflection at their elbows. Scc3^Psc3 and the kleisin middle region bind DNA in a fashion similar to that previously seen in a crystal structure of budding yeast Scc3, with an Scc1 peptide, bound to DNA (*Li et al., 2018*). The DNA interaction is again provided by an array of positively charged amino acids that line

the combined Scc3$^{Psc3}$ and kleisin surface. We refer to this second DNA binding site as the 'Scc3-hinge module' (see *Figure 1—figure supplement 1A* for details of the structural model).

## Predicted conformational changes following ATP hydrolysis

We now consider the consequences of ATP hydrolysis on the two DNA binding modules outlined above. Upon ATP hydrolysis, the ATPase heads disengage, leading to loss of at least some of the DNA contacts within the Scc2-head module. If we assume that Scc2$^{Mis4}$ returns to its extended crystal structure form, further DNA contacts are lost as the gripping state opens up (*Figure 1B*, right). We can see how, in this state, DNA is free to leave the Scc2-head module during topological DNA entry (a structural model for the full DNA trajectory during topological entry is provided in Figure 3).

During loop extrusion, we propose that DNA fails to exit the Scc2-head module through the ATPase head gate following ATP hydrolysis. This could be because an alternative kleisin path obstructs the head gate, or due to persisting electrostatic interactions between Scc2$^{Mis4}$ and the DNA (discussed below in Figure 3). DNA movements are instead limited to transverse DNA sliding, that is in and out of the image plane in *Figure 1B*, akin to experimentally observed cohesin loader sliding along DNA (*Stigler et al., 2016*). Because of the moveable nature of the DNA interaction, we refer to this shape of the Scc2-head module as its 'slipping state'.

A bent Scc2$^{Mis4}$ conformation that embraces DNA, compared to its extended crystal structure form, is shared between the fission yeast, human and budding yeast gripping states (*Figure 1—figure supplement 1B*; *Collier et al., 2020*; *Higashi et al., 2020*; *Shi et al., 2020*). This commonality opens up the possibility that the gripping to slipping state conformational transition is a conserved feature of the Scc2-head module.

In contrast to the Scc2-head module, the Scc3-hinge module and its DNA binding site do not undergo an obvious conformational change when comparing its gripping state and free crystal structure forms. Human Scc3$^{SA1}$ in the gripping state shows an almost perfect overlap with the crystal structure conformation of free Scc3$^{SA2}$ (RMSD = 2.4 Å, *Figure 1—figure supplement 1C*; *Hara et al., 2014*; *Shi et al., 2020*). In the gripping state, Scc3$^{Psc3}$ interacts with the cohesin loader both along the N-terminal Scc2$^{Mis4}$ handle, as well as the central Scc2$^{Mis4}$ hook. Scc2$^{Mis4}$ rearrangement into its extended form disrupts at least some of these contacts, thereby terminating Scc3$^{Psc3}$ - Scc2$^{Mis4}$ juxtaposition (*Figure 1C*, right). A conformational change within the Scc2$^{Mis4}$ handle is furthermore likely to weaken its interaction with the SMC hinge. We therefore hypothesize that, as a consequence of Scc2$^{Mis4}$ structural rearrangements following ATP hydrolysis, the interaction between the Scc2-head and Scc3-hinge modules resolves. While the Scc2-head module turns from the DNA gripping to the slipping state, the DNA binding characteristics of the Scc3-hinge module remain unaltered.

## Measured positional changes between the Scc3-hinge and Scc2-head modules

Above, we predicted positional changes of the Scc3-hinge module relative to the Scc2-head module, when comparing the gripping and ATP post-hydrolysis states. To experimentally observe the positions of module components, we designed FRET reporters inserted at the hinge within Smc1$^{Psm1}$, at the C-terminus of Scc3$^{Psc3}$ and at the N-terminus of Scc2$^{Mis4(N191)}$ (*Figure 2A*). Scc2$^{Mis4(N191)}$ is an N-terminally truncated Scc2$^{Mis4}$ variant missing the first 191 amino acids. The truncation abrogates Scc4$^{Ssl3}$ interaction, a factor important for in vivo cohesin loading onto chromatin. In vitro, using naked DNA as a substrate, Scc2$^{Mis4(N191)}$ retains full biochemical capacity to promote gripping state formation, topological cohesin loading, as well as loop extrusion (*Chao et al., 2015*; *Higashi et al., 2020*; *Shi et al., 2020*). Based on our structural model, these locations are within distances that should allow FRET signal detection in the gripping state. CLIP or SNAP tags, inserted at these positions, served as fluorophore receptors. We labeled these tags during protein purification with Dy547 and Alexa 647 dyes as donor and acceptor fluorophores, respectively (*Figure 2B* and *Figure 2—figure supplement 1A*). The tagged and labeled proteins retained the ability to topologically load onto DNA in vitro, albeit at slightly reduced efficiencies (*Figure 2—figure supplement 1B*). We then mixed labeled cohesin, cohesin loader, a 3 kb circular double stranded plasmid DNA and ATP in the indicated combinations. To create the gripping state, we included all components but substituted ATP for the non-hydrolyzable nucleotide ground state mimetic ADP · BeF$_3$. Following Dy547

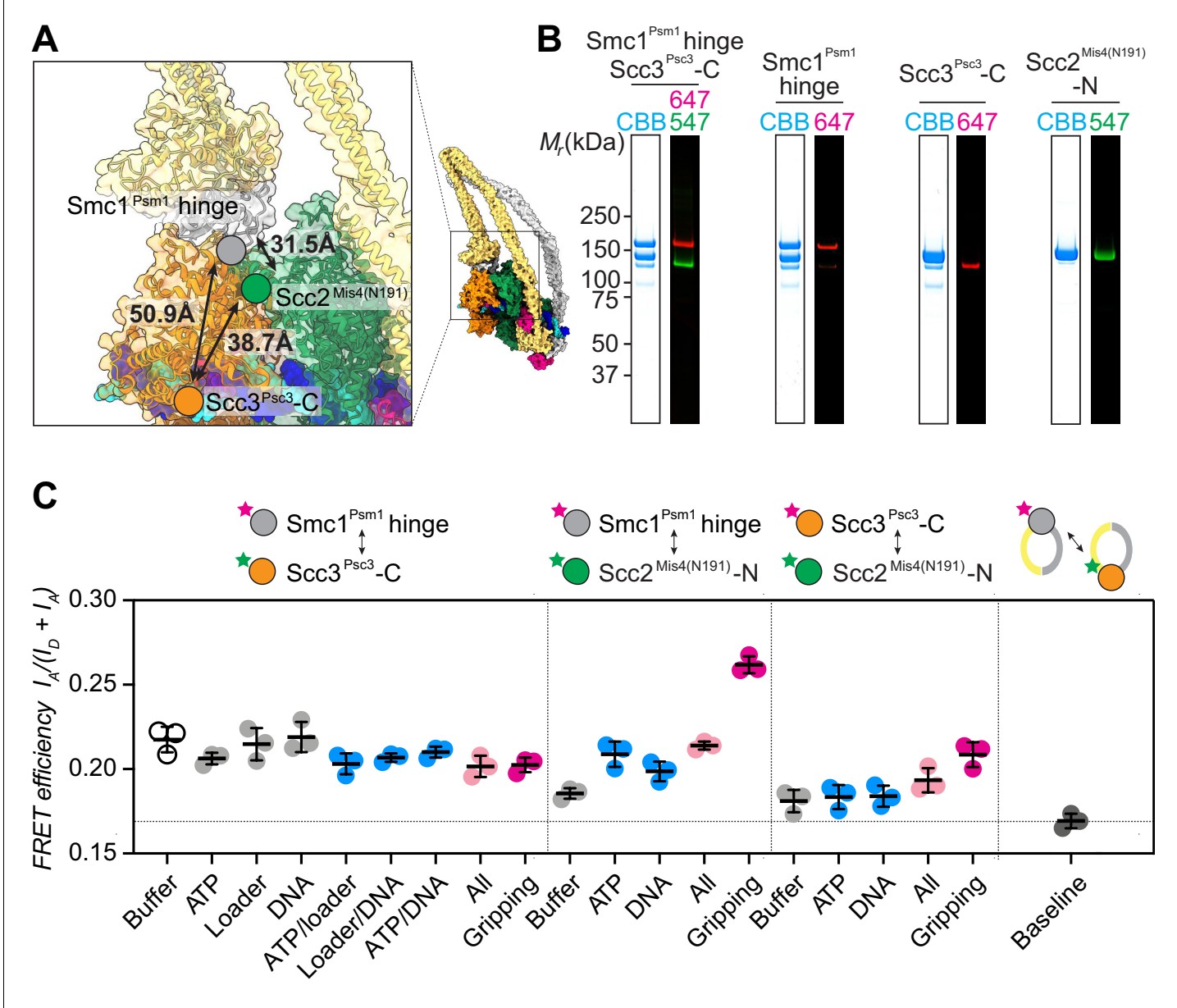

**Figure 2.** FRET-based conformational analyses of the Scc3-hinge and Scc2-head modules. (A) FRET reporter positions on the fission yeast cohesin structural model in the DNA gripping state. The silver, orange, and green circles mark the positions of the Smc1[Psm1] hinge residue R593, the Scc3[Psc3] C-terminal residue E959 and Scc2[Mis4] residue P209, respectively. The Euclidean distances between the $C_\alpha$ atoms of these residues are indicated. (B) The purified and labeled cohesin complexes and Scc2[Mis4(N191)] cohesin loader were analyzed by SDS-PAGE followed by Coomassie blue staining (CBB) or in gel fluorescence detection of the Cy547 (547) and Alexa 647 (647) dyes. (C) Relative FRET efficiencies $I_A/(I_D + I_A)$ between the respective elements were recorded under the indicated conditions, where $I_D$ is the donor and $I_A$ the acceptor emission intensity resulting from donor excitation. The apparent FRET value observed using a mixture of single-labeled cohesins is indicated as a baseline. Results from three independent repeats of the experiments, their means and standard deviations are shown.

The online version of this article includes the following figure supplement(s) for figure 2:

**Figure supplement 1.** Control experiments for the FRET-based conformational analyses.

**Figure supplement 2.** Investigation of complex formation between cohesin and the Scc2[Mis4(N191)] cohesin loader.

excitation, we measured the relative FRET efficiency, defined as the Alexa 647 emission divided by the sum of Dy547 and Alexa 647 emissions.

We first recorded FRET between the fluorophore pair at the Smc1$^{Psm1}$ hinge and the Scc3$^{Psc3}$ C-terminus. The FRET efficiency measured with the cohesin complex alone was 0.22 and displayed only negligible changes following the addition of one or more of the different cofactors. Even under conditions of gripping state formation, the FRET efficiency remained unchanged (*Figure 2C*). As a control, we prepared a mixture of singly Smc1$^{Psm1}$ hinge and singly Scc3$^{Psc3}$ C-terminus labeled cohesin complexes. This mixture provides a baseline for the apparent background FRET value due to spectral overlap. At 0.17 the measurement remained substantially below the FRET values observed when both fluorophores were incorporated within the same cohesin complex. This observation supports the idea that the SMC hinge and Scc3$^{Psc3}$ lie in proximity of each other to form an Scc3-hinge module, consistent with biochemically observed Scc3$^{Psc3}$-hinge binding (*Murayama and Uhlmann, 2015*). Module formation was observed under all tested conditions, irrespective of the stage during cohesin's ATP binding and hydrolysis cycle.

Next, we investigated the positioning of the Scc3-hinge module relative to the Scc2-head module. We first measured FRET between a donor fluorophore at the Scc2$^{Mis4(N191)}$ N-terminus and an acceptor fluorophore at the Smc1$^{Psm1}$ hinge. We observed FRET at relatively low values under most conditions. Strikingly, the FRET efficiency markedly increased under conditions of gripping state formation (*Figure 2C*). This observation confirms that the Scc3-hinge and Scc2-head modules come into proximity in the ATP-bound gripping state, as seen in the cryo-EM structures. We also measured FRET between the Scc2$^{Mis4(N191)}$ N-terminus and an acceptor fluorophore at the Scc3$^{Psc3}$ C-terminus. Again, FRET showed a relative increase under conditions of gripping state formation. The absolute FRET efficiency at this fluorophore pair remained lower when compared with the acceptor fluorophore at the Smc1$^{Psm1}$ hinge. This is expected from a longer predicted Euclidean distance between Scc2$^{Mis4(N191)}$ and the Scc3$^{Psc3}$ C-terminus in the gripping state, compared to Scc2$^{Mis4(N191)}$ and the Smc1$^{Psm1}$ hinge (*Figure 2A*). Together, these observations suggest that the Scc3-hinge and Scc2-head modules come close to each other in the gripping state but separate from each other in other conditions.

When using Scc2$^{Mis4(N191)}$ as the FRET donor, its transitory interaction with cohesin becomes a confounding factor. Higher FRET efficiency in the gripping state could have been due to increased cohesin-Scc2$^{Mis4(N191)}$ complex formation, rather than a conformational change. To examine this possibility, we monitored the cohesin-Scc2$^{Mis4(N191)}$ interaction by co-immunoprecipitation. This revealed equal interaction efficiencies under all of our incubation conditions (*Figure 2—figure supplement 2*). Therefore, the observed FRET differences cannot be explained by different cohesin-Scc2$^{Mis4(N191)}$ complex stabilities. Rather, the FRET changes indeed point to conformational transitions within the cohesin complex.

## The role of the Scc3-hinge module during topological DNA entry

What are the consequences of the Scc3-hinge module, and its movement relative to the Scc2-head module, for the DNA trajectory during topological DNA entry? Our earlier results suggested that DNA arrives from the top of the ATPase heads and usually passes the kleisin N-gate before reaching the gripping state (*Figure 3A*, panel *a*) (*Higashi et al., 2020*). The kleisin N-gate initially opens as the consequence of ATP-dependent SMC head engagement (*Muir et al., 2020*). A positively charged kleisin N-tail then guides DNA through this gate *en route* to the gripping state. In the gripping state, the DNA together with the Scc2$^{Mis4}$ cohesin loader shut the gate, while the Scc3-hinge module docks onto the Scc2-head module. The straight DNA path through both DNA binding modules in turn requires that the DNA bends where it arrives between the Smc1$^{Psm1}$ and Smc3$^{Psm3}$ coiled coils. The DNA path shown in *Figure 3A*, panel *a*, highlights the position of the bend, based on our DNA-protein crosslink mass spectrometry data (*Higashi et al., 2020*). The notion of DNA bending in the gripping state finds further support from magnetic tweezer experiments, in which condensin introduced a discrete DNA shortening step under gripping state conditions (*Ryu et al., 2020b*).

A stable DNA gripping state forms only in the presence of non-hydrolyzable ATP. Usually, gripping state formation triggers ATP hydrolysis, resulting in ATPase head gate opening and Scc3-hinge and Scc2-head module uncoupling. This uncoupling allows a swinging motion of the Scc3-hinge module and proximal coiled coil, with a pivot point at the elbow (*Figure 3A*, panel *b*). No force needs to be transmitted along the SMC coiled coil for this swinging motion to initiate. Rather,

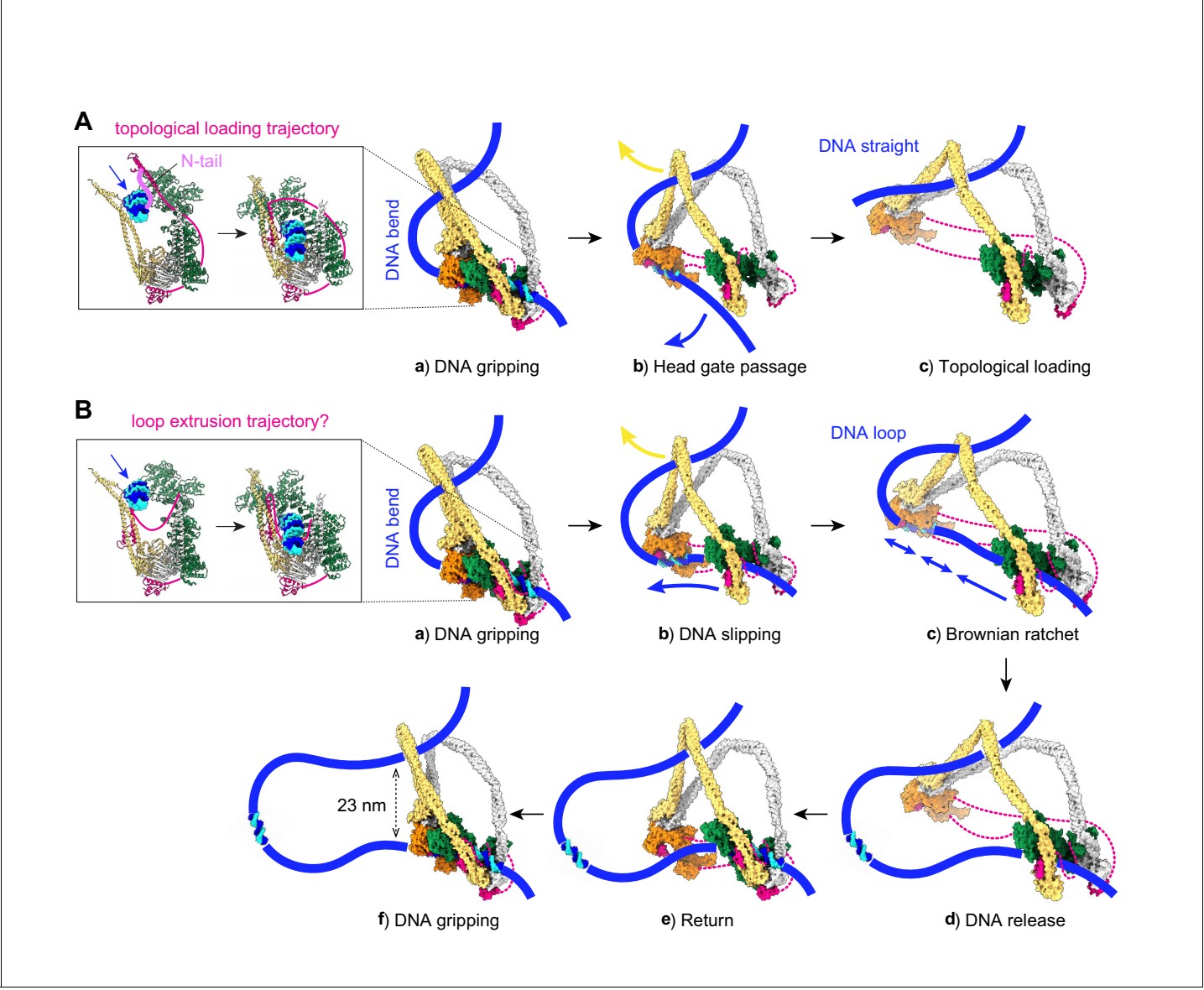

**Figure 3.** Molecular models for topological DNA entry into the cohesin ring and for loop extrusion. (**A**) Topological DNA entry into the cohesin ring. (**a**) DNA enters through the open kleisin-N gate. DNA binding by both the Scc2-head and Scc3-hinge modules introduces a DNA bend. (**b**) ATP hydrolysis opens the SMC head gate. The swinging motion of the Scc3-hinge module (indicated by a yellow arrow) steers the DNA through the head gate to complete topological entry. (**c**) DNA is topologically entrapped in the Scc3-Smc3-kleisin-N chamber. (**B**) Loop extrusion. (**a**) DNA arrives in the gripping state without passing the kleisin N-gate. (**b**) ATP hydrolysis leads to SMC head gate opening, but the kleisin path prevents DNA passage. The swinging motion of the Scc3-hinge module instead turns the DNA bend into a loop, while DNA slips along the Scc2-head module. (**c**) Loop growth depends on the stochastic Brownian motion of the Scc3-hinge module. (**d**) The low DNA affinity of the Scc3-hinge module results in DNA release. (**e**) The Scc3-hinge module returns to form a new DNA gripping state upon nucleotide binding. (**f**) The next loop extrusion cycle begins. The in- and outbound DNAs are constrained by cohesin at a distance of ~23 nm, in line with recent measurements of the condensin neck size when engaged in loop extrusion (*Ryu et al., 2020a*).

The online version of this article includes the following figure supplement(s) for figure 3:

**Figure supplement 1.** Additional views of cohesin during topological loading onto DNA and during loop extrusion.

following release, Brownian motion can take the Scc3-hinge module only in one direction, away from the Scc2-head module. When we consider the consequence of the Scc3-hinge swinging motion on the DNA path, we make two observations. Firstly, the bent DNA straightens, an effect that might

favor the swinging motion. Secondly, the movement effectively steers the DNA through the ATPase head gate to complete topological entry into the cohesin ring.

Following head gate passage, we expect that DNA retains Scc3-hinge module association only for a limited time. DNA affinity to Scc3$^{Psc3}$ and the kleisin middle region in isolation has been measured at around 2 μM (*Li et al., 2018*), a relatively low affinity that implies a fast off-rate once Scc3$^{Psc3}$ has left the gripping state. DNA consequently finds itself in a cohesin chamber delineated by Scc3$^{Psc3}$, the Smc3$^{Psm3}$ coiled coil, as well as the unstructured part of the kleisin between the kleisin N-gate and the kleisin middle region (*Figure 3A*, panel *c*). We refer to this space as cohesin's Scc3-Smc3-kleisin-N chamber. Two separase recognition sites in Scc1$^{Rad21}$, whose cleavage liberates DNA from the cohesin ring to trigger anaphase (*Tomonaga et al., 2000*; *Uhlmann et al., 2000*), are situated within this part of the kleisin unstructured region (*Figure 3—figure supplement 1A*).

Single molecule imaging of cohesin, topologically loaded onto DNA, showed that its diffusion is blocked by obstacles smaller than those expected to be accommodated by cohesin's SMC compartment (*Davidson et al., 2016*; *Kanke et al., 2016*; *Stigler et al., 2016*). This observation is consistent with the possibility that DNA resides in a sub-chamber of the cohesin ring following topological loading. How durable the Scc3$^{Psc3}$-SMC hinge association is, whether DNA permanently resides inside the Scc3-Smc3-kleisin-N chamber, or whether subunit rearrangements take place following successful topological loading, for example when the cohesin loader is replaced by Pds5, remains to be further ascertained.

## An alternative gripping state that initiates loop extrusion

The structured components of the gripping state do not by themselves contain information about whether DNA has in fact passed the kleisin N-gate. While mechanisms are in place to ensure kleisin N-gate passage, for example the kleisin N-tail, DNA might under certain conditions reach the gripping state without having passed this gate (*Figure 3B*, panel *a*). What will be the consequence of ATP hydrolysis in such an alternative gripping state? The Scc2-head module turns into its DNA slipping state, but the kleisin path prevents DNA from passing between the ATPase heads. The Scc3-hinge module again uncouples from the Scc2-head module, but now its diffusion-driven swinging motion cannot steer DNA through the head gate. The only way for the Scc3-hinge module to launch its swinging motion is to further bend the DNA, turning it into a loop, while DNA slips through the Scc2-head module (*Figure 3B*, panel *b*). The directed diffusive motion of the Scc3-hinge module has created a Brownian ratchet, allowing DNA motion only in one direction (*Figure 3B*, panel *c*). The entropy gain from gripping state disassembly in turn helps to offset the energetic cost of DNA loop formation.

Once a DNA loop is initiated, the extent of loop growth per reaction cycle is limited by how far the Scc3-hinge and Scc2-head modules separate from each other. The maximum separation is likely dictated by the kleisin unstructured regions that link Scc3$^{Psc3}$ to the Scc2-head module. Their lengths of 135 amino acids (between the Scc2$^{Mis4}$ and Scc3$^{Psc3}$ binding sites) and 109 amino acids (between the Scc3$^{Psc3}$ binding site and the kleisin C-terminal domain) gives a conservative estimate of ~40 nm (*Figure 3—figure supplement 1B*; *Ainavarapu et al., 2007*). This distance allows considerable, but perhaps not complete, extension of the ~47 nm long SMC proteins. As we will see below, the actual amount of loop growth is likely less and depends on the SMC elbow angle reached by stochastic diffusive motion at the time when DNA dissociates (*Figure 3B*, panel *d*).

After DNA dissociation from the Scc3-hinge module, there is a time when there is only loose cohesin-DNA contact with the Scc2-head module. Thermal fluctuations now lead to random loop size changes, depending on the probability of diffusion and on external forces that might apply. As long as the Scc2$^{Mis4}$ cohesin loader remains part of the Scc2-head module, the local proximity of all components means that a return of the Scc3-hinge module (*Figure 3B*, panel *e*) and the establishment of a new DNA gripping state following nucleotide binding (*Figure 3B*, panel *f*) are very likely events. The next loop extrusion cycle begins.

## Variants of the molecular DNA loop extrusion model

The above model makes a specific molecular proposal for DNA loop extrusion by the cohesin complex. At its core lies a Brownian ratchet built from the Scc3-hinge and Scc2-head modules, juxtaposed during gripping state formation but allowing unidirectional DNA diffusion following ATP

hydrolysis. This robust core of a loop extrusion mechanism tolerates variations in its surrounding features.

One point of uncertainty regards the Scc3$^{Psc3}$ interaction with the SMC hinge. While our FRET observations suggest proximity of Scc3$^{Psc3}$ and the SMC hinge under all tested solution conditions, others have observed Scc3$^{Psc3}$ close to the ATPase heads and distant from the hinge (*Anderson et al., 2002*; *Huis in 't Veld et al., 2014*). It is possible that the Scc3$^{Psc3}$-hinge interaction has a limited lifetime following gripping state disassembly. As long as Scc3$^{Psc3}$ retains hinge association during initial SMC coiled coil unfolding, the Brownian ratchet has served its purpose. Scc3$^{Psc3}$ might then dissociate from both the DNA and the hinge without impacting on loop extrusion efficiency (*Figure 3—figure supplement 1C*), as long as all subunits rejoin during gripping state reassembly.

Another open question in our molecular proposal is whether DNA indeed fails to pass the kleisin N-gate before the initiation of loop extrusion. The finding that loop extrusion is not prevented by covalent fusions between cohesin ring subunits (*Davidson et al., 2019*) does not strictly answer the question whether or not DNA passes the kleisin N-gate, which lies at a distinct distance from the kleisin N-terminus. If DNA did pass the kleisin N-gate, as it would during topological entry into the cohesin ring, the kleisin path would not obstruct DNA exit from the Scc2-head module. Instead, under the low-salt conditions typical for loop extrusion experiments, DNA might retain electrostatic contact with the positive charges of the Scc2-head module in the slipping state (*Figure 3—figure supplement 1D*). Whether or not such interactions are sufficient to sequester DNA within the Scc2-head module during loop extrusion remains to be learned.

## Loop extrusion by the fission yeast cohesin complex

The above molecular model for loop extrusion is based on experiments using the fission yeast cohesin complex and its loader. To date, loop extrusion by cohesin has only been observed using human proteins, while attempts to observe loop extrusion by budding yeast cohesin have remained unsuccessful (*Ryu et al., 2021*). We therefore asked whether fission yeast cohesin is able to extrude DNA loops. We employed a single-molecule assay similar to those previously used to observe DNA loop extrusion by budding yeast condensin and human cohesin (*Ganji et al., 2018*; *Davidson et al., 2019*). Individual molecules of λ-phage DNA were tethered via both ends to a cover glass surface of a microfluidic flow cell, stained with Sytox Orange and stretched by continuous buffer flow while being imaged using total internal reflection (TIRF) microscopy. In the presence of 5 nM fission yeast cohesin, the cohesin loader and ATP, over 40% of the DNA molecules were seen to display active DNA loop extrusion (*Figure 4A*, *Video 1*). Loop extrusion proceeded in a symmetrical manner at a mean rate of ~1 kbp s$^{-1}$ (maximal rate 2.4 kbp s$^{-1}$, *Figure 4B*) resembling DNA loop extrusion by human cohesin (*Davidson et al., 2019*; *Kim et al., 2019*). This confirms that fission yeast cohesin can both, topologically load onto DNA and extrude DNA loops.

## A computational model for Brownian ratchet-driven DNA loop extrusion

In our proposed model of loop extrusion, two DNA binding modules within the cohesin complex generate a Brownian ratchet. The ratchet is operated by repeated cycles of ATP-dependent DNA gripping state formation and its unidirectional dissolution following ATP hydrolysis. To evaluate whether such a mechanism is physically plausible, we constructed a structure-based molecular-mechanical model of the cohesin-DNA interaction and carried out computational simulations to explore its behavior. We modeled DNA as a discrete stretchable, shearable wormlike chain (dssWLC), which describes DNA with persistence length as the only parameter (*Figure 5A*; *Koslover and Spakowitz, 2014*). We assumed the persistence length to be $L_p$ = 50 nm (*Wang et al., 1997*; *Bustamante et al., 2000*). The cohesin coiled coil segments as well as the linkage between the two SMC heads were modeled using the same approach. Each coiled coil was represented as three beads that interact via a dssWLC (*Figure 5B*). This again leaves persistence length as the sole parameter that we chose such that it leads to a head-to-hinge distance distribution that matches experimentally measured head-to-hinge distances in a freely diffusing eukaryotic SMC complex (*Ryu et al., 2020a*).

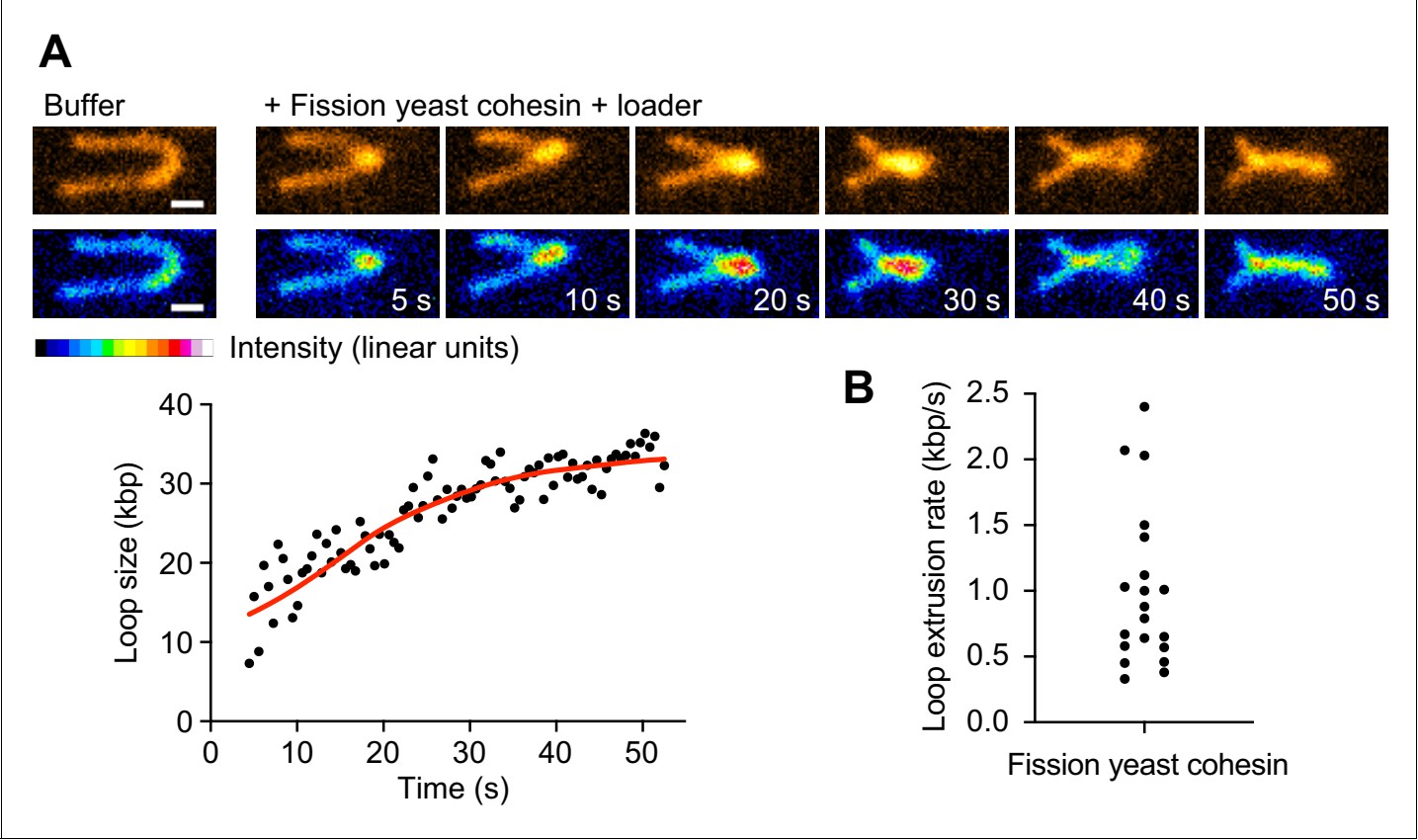

**Figure 4.** Loop extrusion by the fission yeast cohesin complex. (**A**) Time course of DNA loop extrusion. Scale bar 1 μm. A doubly tethered DNA molecule (48.5 kbp) is stained with Sytox Orange and stretched using buffer flow at 15 μl/min. Upon addition of cohesin and cohesin loader in the presence of ATP, a loop formed at the tip of the DNA and dynamically extended toward the DNA ends. The corresponding loop size change over time is plotted. (**B**) Mean DNA loop extrusion rates from $n$ = 21 DNA molecules analyzed.

Based on our structural and biochemical observations, we define two states of the cohesin complex. In the gripping state, the Scc3-hinge and Scc2-head modules are engaged and the coiled coil elbows are folded. Both modules make stable contact with DNA (*Figure 5C*, Gripping state). In the second state, the slipping state, the Scc3-hinge and Scc2-head modules do not interact, allowing an unfolded cohesin conformation. In this state, the Scc2-head module permits free transverse DNA motion. DNA association with the Scc3-hinge module, defined by its equilibrium dissociation constant, is manually controlled in our model (*Figure 5C*, Slipping state).

First, we explored the dynamics of the transition between cohesin's gripping and slipping states. As a starting point we assume that a small DNA loop is inserted into the cohesin ring in the gripping state. The first panel in *Figure 5D* shows a snapshot of this initial state after equilibration by the Metropolis Monte-Carlo algorithm (see Materials and methods). Our 3D model has no explicit chemical kinetics, and to simulate transitions between chemical states we prescribed parameter changes corresponding to a new state, then sampled a sufficient number of iterations to reach a new equilibrium. To simulate transition to the slipping state, we impose parameter changes that

**Video 1.** Movie showing formation and extension of a loop on doubly tethered DNA in the presence of cohesin, the cohesin loader and ATP. Scale bar 1 μm. A DNA molecule (48.5 kbp) is stained with Sytox Orange and stretched at 15 μL/min buffer flow.
https://elifesciences.org/articles/67530#video1

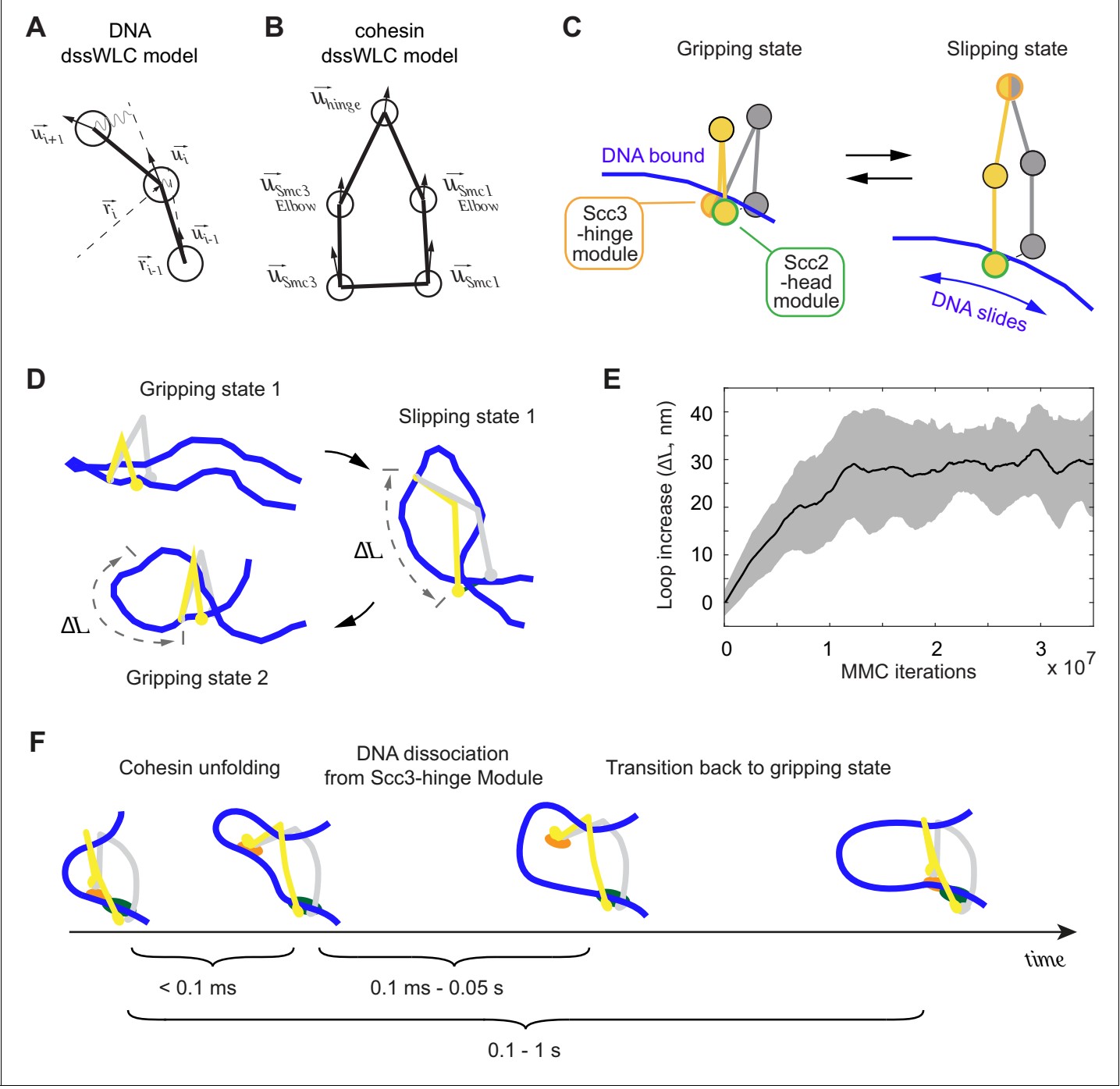

**Figure 5.** 3D Metropolis Monte-Carlo simulation of cohesin-DNA interactions. (A) Representation of DNA as a dssWLC model. DNA is split into 5 nm long segments. Each segment is described by its radius vector $\vec{r}_i$ and a unit vector $\vec{u}_i$ that defines segment orientation. (B) Cohesin is described by five beads corresponding to Smc1 and Smc3 heads and elbows as well as the hinge. The bead positions are defined by the corresponding vector radii (not shown) and their orientation by unit vectors. (C) The two equilibrium conformations of the model, corresponding to the gripping and slipping states. (D) Snapshots from a simulation, started in a gripping state. The system was sampled $2.5 \cdot 10^6$ times and then the equilibrium state was changed to the slipping state. It was sampled for another $8 \cdot 10^6$ iterations, which led to cohesin unfolding and loop extension. At iteration $1 \cdot 10^7$, the equilibrium conformation was returned to gripping state. (E) Loop length increase shown as function of Metropolis Monte-Carlo (MMC) iterations following the gripping to the slipping state transition. Before iteration zero, the system was equilibrated for $2 \cdot 10^6$ rounds in the gripping state. The black line represents the average and the gray area the standard deviation across 10 independent simulation replicates. (F) Schematic to show time progression of the DNA interaction with the Scc3-hinge module. Time intervals show indicative ranges of model parameters.

disconnect the Scc3-hinge from the Scc2-head module and switch the Scc2-head module to its slipping state, while the Scc3-hinge module remains bound to DNA. We then sampled conformations with the new parameters until a new equilibrium was reached. As cohesin unfolds, DNA binding to the Scc3-hinge module limits DNA movement at the Scc2-head module to only one direction, toward an increased loop size (*Figure 5D*). The average increase in loop size during multiple repeats of this transition is ~30 nm (*Figure 5E*). When we then prescribe DNA detachment from the Scc3-hinge module and switch cohesin back to the gripping state, the system readily resets and primes itself for the next cycle (*Figure 5D*). Our simulations reveal that repeated rounds of the states: 'gripping - > slipping - > DNA detachment from the Scc3-hinge module - > gripping' results in continuous extrusion of DNA with an average loop size increase of ~30 nm per cycle (*Video 2*).

## DNA affinity of the Scc3-hinge module controls loop extrusion

In the above computational model, all state transitions were manually prescribed to obtain cycles of DNA loop extrusion. In the following, we consider requirements for such cycles to arise based on the chemical kinetics of the cohesin complex. We assume, based on the structural data, that both the Scc2-head and Scc3-hinge modules bind DNA in the ATP-bound gripping state. Following ATP hydrolysis, the Scc2-head module switches to its slipping state, while DNA remains bound to the Scc3-hinge module. To achieve processive cycles of loop extrusion, DNA binding must persist for long enough to ensure biased DNA diffusion toward loop growth while cohesin unfolds (*Figure 5F*). An upper limit for the time it takes cohesin to unfold is given by the time of a diffusive process that separates the Scc3-hinge and Scc2-head modules. Assuming molecular masses of both modules in the 200 kDa range, it takes ~0.1 ms for them to diffuse ~50 nm apart (see Materials and methods). This time is an upper estimate. If cohesin opening was driven not merely by diffusion, but assisted by internal stiffnesses of the coiled coils, this could speed up opening. Based on this estimate, our model predicts that the DNA off-rate at the Scc3-hinge module should be lower than $1/0.1$ ms = 10,000 $s^{-1}$ in order for DNA to maintain Scc3-hinge module association until cohesin fully unfolds.

After cohesin has opened, two further scenarios are possible. Firstly, DNA could dissociate from the Scc3-hinge module before cohesin transitions back into the next gripping state. In this case a loop length gain is made and the ensuing gripping state primes cohesin for the next round of loop extrusion (*Figure 5F*). Alternatively, cohesin could switch back to the gripping state before DNA is released from the Scc3-hinge module. In this situation, DNA ends up in the same position as before and there is no net loop size gain, resulting in an unproductive cycle. Based on these considerations, loop extrusion in our model is most effective when the DNA lifetime at the Scc3-hinge module is longer than the time required for diffusion-driven cohesin unfolding, but shorter than it takes cohesin to transition back into the next gripping state. Such a lifetime would ensure that most reaction cycles result in net loop growth.

The ATP-bound DNA gripping state is an unstable state, so we can expect cohesin to spend the majority of its time in the post-hydrolysis slipping state. We can then approximate the lifetime of the slipping state based on cohesin's ATP hydrolysis rate. This rate has been measured with a lower limit of ~ 2 $s^{-1}$ (*Davidson et al., 2019*; *Ganji et al., 2018*; *Murayama and Uhlmann, 2014*). As two ATPs are coordinately hydrolyzed by the two ATPase heads, this equates to a cycle rate of ~ 1 $s^{-1}$.

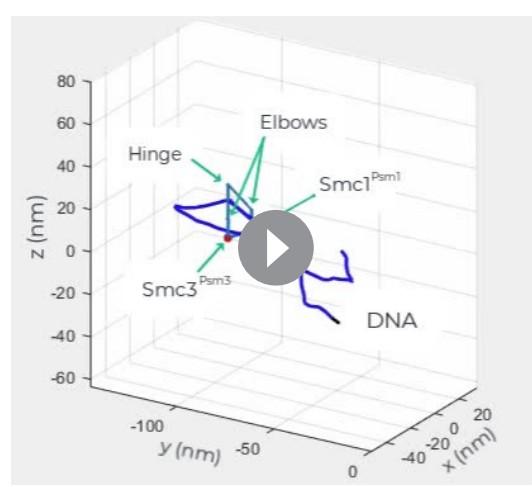

**Video 2.** Progression of the Metropolis Monte-Carlo simulation as cohesin is switched between the parameters describing its gripping and slipping states (*Supplementary file 1*). Before the start of the video, the system was equilibrated for ~ $10^7$ iterations. Frames were captured every $2 \cdot 10^5$ iterations and the video spans ~ $5 \cdot 10^7$ iterations. Pauses were introduced to highlight imposed state transitions and DNA binding and unbinding events. Axis scale, nm.
https://elifesciences.org/articles/67530#video2

Thus, our model predicts that efficient loop extrusion is achieved when the DNA off-rate from the Scc3-hinge module is in the range of 1 – 10,000 s$^{-1}$. While we do not know the actual off-rate, the equilibrium dissociation constant $K_d$ between DNA and Scc3 has been measured at ~ 2 µM (*Li et al., 2018*). Assuming an association rate $k_{on}$ typical for biomolecular interactions of ~ 10$^7$ M$^{-1}$ s$^{-1}$ (*Howard, 2001*), we arrive at a corresponding off-rate $k_{off} = K_d \times k_{on}$ of ~ 20 s$^{-1}$. This value sits well within the range predicted to support processive loop extrusion. The number ensures processivity even if cohesins that are actively engaged in loop extrusion undergo conformational cycles and hydrolyze ATP up to twenty times faster than measured in bulk solution experiments.

### Loop extrusion by biased DNA diffusion

Having established that transitions between cohesin's gripping and slipping states can drive directional DNA movements, we explored how this mechanism compares to available experimental observations of loop extrusion at realistic time scales. To do this, we constructed a simplified model of loop development. We assume that both DNAs that enter cohesin at the Scc2-head module and exit cohesin at the Smc3$^{Psm3}$ elbow can randomly diffuse in and out of the ring with rates depending on a DNA diffusion coefficient $D$. We then use a Monte-Carlo algorithm to simulate DNA loop dynamics as a function of time.

If we adopt a diffusion coefficient of ~1 µm$^2$/s, as measured for cohesin movements on DNA following topological loading (*Davidson et al., 2016*; *Kanke et al., 2016*; *Stigler et al., 2016*), we see that both strands randomly diffuse back and forth, leading to stochastic loop size changes (*Figure 6A*). Within a few minutes, a typical time frame used to microscopically observe DNA loop extrusion, the loop size changes over a range of several kilobases. However, these random movements do not show a preferred direction and cannot drive loop extrusion.

This situation changes when the Scc3-hinge module engages with DNA in the gripping state and disengages predominantly in the slipping state. The Scc3-hinge module restricts DNA diffusion at the Scc2-head module to only one direction – toward loop growth. This effect applies only to the DNA that enters the loop at the Scc2-head module, but not to the DNA that exits cohesin. We assume that the latter DNA continues to diffuse randomly in both directions irrespective of the cohesin state. If we simulate directed DNA motion at the Scc2-head module of 30 nm per cohesin turnover cycle, we see how, overlaid over stochastic diffusive loop size fluctuations, the loop length steadily increases over time (*Figure 6B*).

We next explored how the variables in this model affect the outcome of loop extrusion. There are three independent variables: the two diffusion coefficients that describe the two DNAs that enter and exit cohesin, as well as the ATPase turnover rate, that is the lifetime of each slipping state. We simulated multiple 10 min intervals of cohesin-DNA dynamics and compared loop extrusion rates extracted from these simulations to those determined in our and in published experiments (*Davidson et al., 2019*).

The simulations revealed that the average loop extrusion rate is unaffected by the DNA diffusion coefficients (*Figure 6C*). Indeed, thermal movement of DNA has no net direction and therefore should not contribute to directed loop growth. Instead, the average loop extrusion rate $v$ is simply a product of the step size during cohesin's state transitions $L$ and the frequency $\gamma$ of these events:

$$v = \gamma * L \qquad (1)$$

Using the value of $L$ = 30 nm = 0.088 kb, we find that there must be around 10 successful cohesin state transitions per second to reach experimentally observed average loop extrusion speeds of ~ 1 kb s$^{-1}$. The required rate of cohesin state transitions necessitates an equal rate of ATPase cycles. This means that a cohesin complex that is actively engaged in loop extrusion hydrolyzes ATP ~ 10 times faster than average bulk solution ATP hydrolysis rates suggested.

A striking feature of experimentally observed loop extrusion is a high variation in loop growth rates (*Figure 4B*; *Ganji et al., 2018*; *Davidson et al., 2019*). To obtain insight into the origin of these variations, we compared scatter in our modeled traces with experimental data. We quantified the scatter as the interquartile range, that is the range that contains 50% of datapoints around the median. This analysis revealed that extrusion rate variations strongly correlated with the DNA diffusion coefficient. The bigger the diffusion coefficient, the greater is the variation (*Figure 6D*).

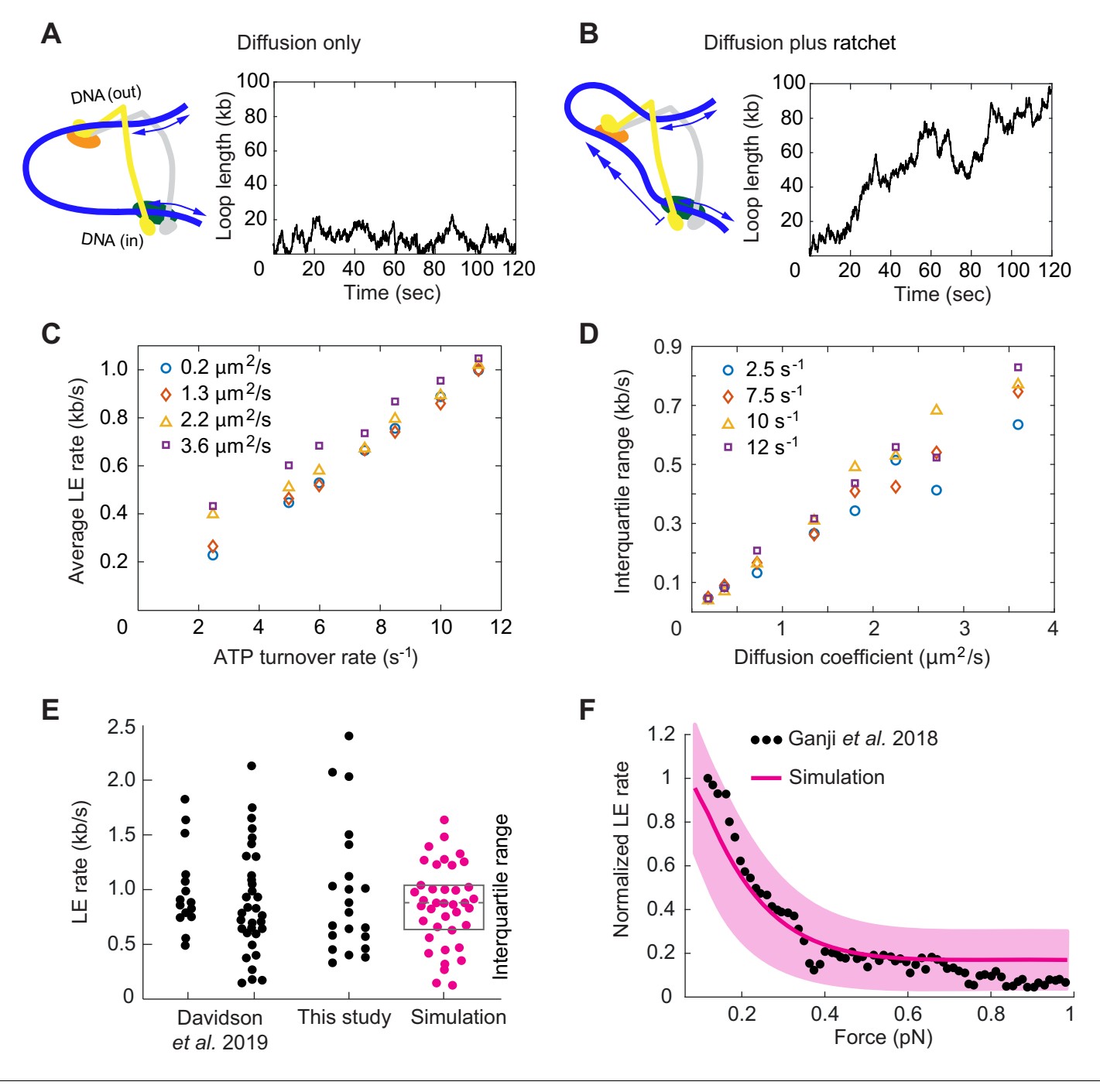

**Figure 6.** Loop extrusion by cohesin as a Brownian ratchet. (A) DNA loop size changes due to diffusion only of both DNAs. (B) DNA loop size changes due to random diffusion plus directed diffusion generated by cohesin's gripping to slipping state transition at a rate of 9 s$^{-1}$. (C) Simulated average loop extrusion (LE) rates as a function of the cohesin cycle (ATP turnover) rate. Symbol colors represent simulations with the indicated diffusion coefficients. (D) Scatter in the loop extrusion rates, generated during 3 min simulations and quantified as the interquartile range, shown as a function of the outbound DNA diffusion coefficient. The diffusion coefficient for the inbound strand was 0.05 µm$^2$/s. Symbol colors indicate simulations with different cohesin cycle rates. (E) Comparison of experimentally observed LE rate distributions of HeLa cell and recombinant human cohesin (left and right, **Davidson et al., 2019**), fission yeast cohesin from **Figure 4**, and an example of simulated data over the same time interval. (F) Comparison of the experimentally observed force dependence of condensin LE rates (**Ganji et al., 2018**) with the simulated outcome. The pink line shows the median across 35 simulations performed with discrete force values at 0.1 pN increments. The light pink area shows the corresponding interquartile range. Parameters for the simulations in (E) and (F) were: cohesin cycle rate = 9 s$^{-1}$, DNA diffusion coefficient = 1.5 µm$^2$/s, simulation time = 3 min. The online version of this article includes the following figure supplement(s) for figure 6:

*Figure 6 continued on next page*

*Figure 6 continued*

**Figure supplement 1.** Interquartile range of loop extrusion (LE) rates when the DNAs entering and exiting the cohesin ring show divergent diffusion coefficients.

Of the two DNAs that enter and exit cohesin, additional simulations showed that only the DNA with the higher diffusion coefficient determines the amount of scatter in extrusion speed (*Figure 6—figure supplement 1*). A diffusion coefficient of ~1.5 µm$^2$/s resulted in a good match to the experimentally observed variation (*Figure 6E*), matching the upper range of experimentally measured values (*Davidson et al., 2016*; *Kanke et al., 2016*; *Stigler et al., 2016*). We imagine that the outward pointing DNA, which is not constrained by the Scc2-head module, might not interact strongly with cohesin and show the greater diffusion coefficient amongst the two DNAs. In addition to the high variations of loop extrusion rates, the low friction of the outward pointing DNA could also explain why DNA can be readily pulled from a condensin complex undergoing loop extrusion (*Kim et al., 2020*).

Finally, we explored how Brownian ratchet-driven loop extrusion in our model is affected by external force. If cohesin unfolding in the slipping state is driven by thermal motion, its rate $k$ in response to external force is given by:

$$k = k_o e^{-\frac{\delta \cdot F}{k_B T}} \tag{2}$$

where $k_0$ is the rate in the absence of external force, $F$ is the external force and $\delta = 30$ nm from our simulations (*Howard, 2001*). Introducing this dependency into our model, we find good agreement between simulations in the presence of a range of applied external forces and the experimentally observed decay of the force-velocity relationship (*Figure 6F*; *Ganji et al., 2018*). Both the similarity between the simulated and experimentally observed responses to external force, as well as the high variation of experimentally observed loop extrusion rates, support the idea of a largely diffusion-driven molecular mechanism of loop extrusion.

## Symmetric loop extrusion as a special case of asymmetric loop extrusion

In our molecular model of DNA loop extrusion, the Brownian ratchet acts only on the DNA that enters the cohesin ring through the Scc2-head module. No directional effect is exerted on the DNA that exits cohesin. This results in asymmetric loop extrusion (*Figure 7A*, Asymmetric loop extrusion), a scenario that is seen in the case of the condensin complex (*Ganji et al., 2018*; *Golfier et al., 2020*). In contrast, both our and the published experimental observations (*Davidson et al., 2019*; *Kim et al., 2019*) suggest that the cohesin complex symmetrically extrudes DNA loops. How could this difference be explained?

In our model, the cohesin loader is a stable part of the cohesin complex. However, Scc2$^{Mis4}$ only weakly binds to the cohesin complex. Suggestive of subunit turnover, the continuous presence of cohesin loader in the incubation buffer is a requirement for processive loop extrusion by human cohesin (*Davidson et al., 2019*). If we picture a situation in which Scc2$^{Mis4}$ dissociates from the cohesin complex, DNA will be released from the Scc2-head module (*Figure 7A*, Symmetric loop extrusion). The DNAs that enter and exit the cohesin ring are now indistinguishable and, once Scc2$^{Mis4}$ rebinds, both DNAs have an equal chance to associate with the Scc2-head module during gripping state formation. Every round of cohesin loader dissociation and reloading thereby results in a one-in-two chance that the extruded DNA strand switches. Averaged over time, this takes the appearance of symmetric loop extrusion.

## Loop extrusion versus directional cohesin translocation

In addition to loop extrusion, single-molecule studies have reported ATP-dependent unidirectional cohesin and condensin translocation along DNA (*Terakawa et al., 2017*; *Davidson et al., 2019*). The experimental conditions under which both complexes move along DNA, or extrude DNA loops, are largely similar. A difference lies in the DNA substrates on which translocation was observed. These substrates were stretched, either by liquid flow or by being double tethered to a flow cell surface. We have seen above that a Brownian ratchet is able to extrude DNA loops only against very

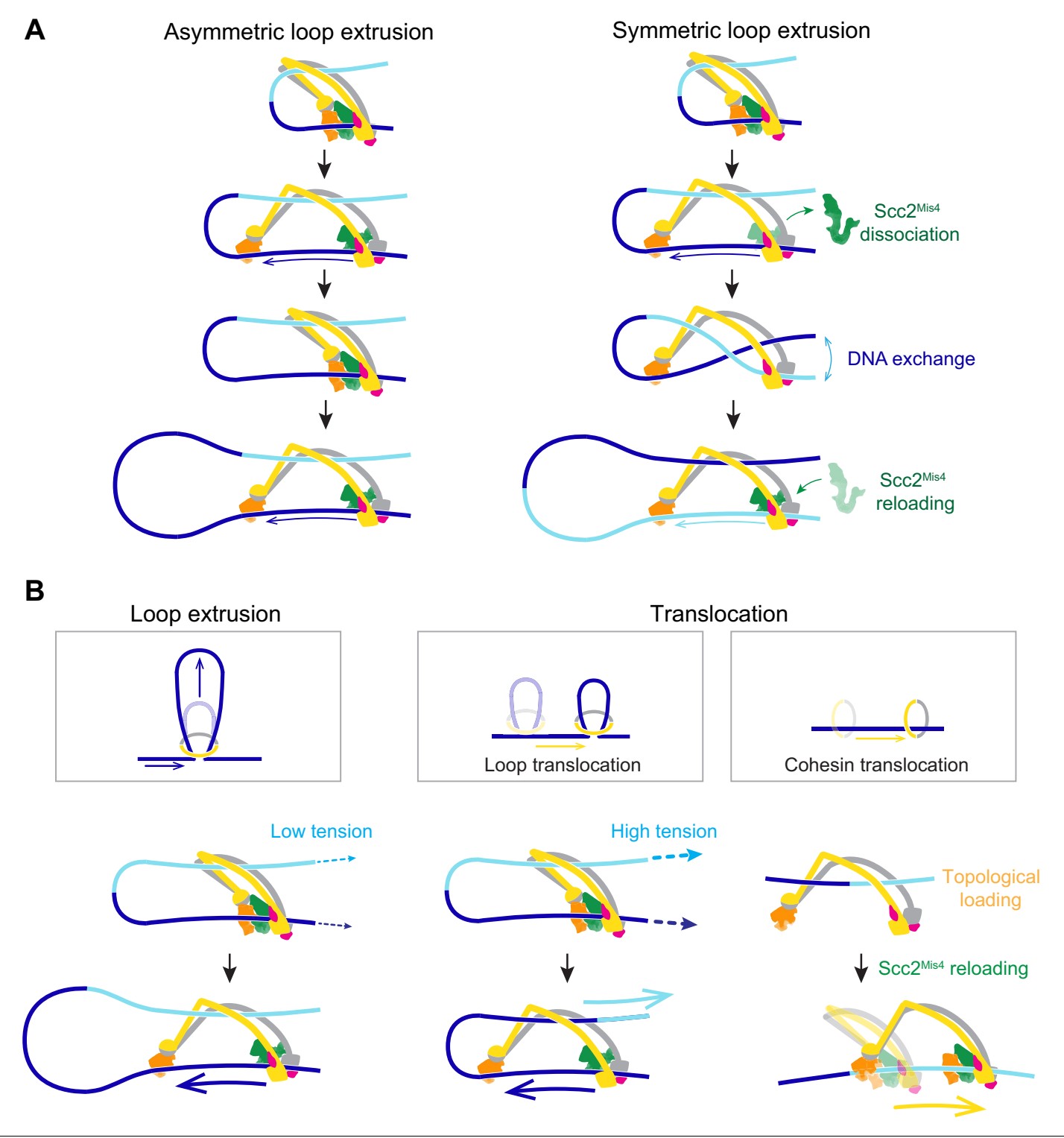

**Figure 7.** A model for asymmetric and symmetric loop extrusion, as well as possible mechanisms for cohesin translocation along stretched DNA. (**A**) A model for symmetric loop extrusion as a special case of asymmetric loop extrusion. Continued action of the Brownian ratchet on the DNA that enters the cohesin ring results in *Asymmetric loop extrusion*. If the cohesin loader dissociates, reassembly of the Brownian ratchet following Scc2[Mis4] reloading gives both DNAs an equal chance to become part of the ratchet. Over time, alternating asymmetric extrusion of both DNAs results in apparent *Symmetric loop extrusion*. (**B**) Two possible mechanisms for cohesin translocation along DNA under tension. *Loop extrusion* is possible only against very weak DNA counterforces. If loop growth stalls due to DNA tension, the continued operation of the Brownian ratchet leads to *Loop translocation*. *Figure 7 continued on next page*

*Figure 7 continued*

Alternatively, cohesin might at first topologically load onto DNA. Recurring gripping state formation and resolution following topological loading results in Brownian ratchet driven *Cohesin translocation*.

small externally applied forces (*Figure 7B*, Loop extrusion). If DNA is longitudinally stretched, loop extrusion will thus be limited to a small loop size. As the Brownian ratchet continues to deliver DNA to enlarge the loop, the stretching force begins to extract the DNA from the opposite side. This will especially be the case if, as suggested above, the diffusion coefficient of the outward pointing DNA is higher than that of the inward moving DNA (*Figure 7B*, Loop translocation). Instead of promoting loop growth, the Brownian ratchet now fuels a motor that moves along the DNA. Consistent with this scenario, a small DNA loop was observed to precede initiation of condensin translocation along stretched DNA (*Ganji et al., 2018*).

We can imagine an alternative scenario by which cohesin could turn into a Brownian ratchet-driven motor. Following successful topological loading onto DNA, cohesin might be able to return to a gripping state, for example if Scc2$^{Mis4}$ is not replaced by Pds5. This scenario finds support from the observation that the cohesin loader retains the ability to stimulate cohesin's ATPase following completion of topological loading (*Çamdere et al., 2015*). Repeated gripping to slipping state transitions could then result in cohesin translocation along DNA (*Figure 7B*, Cohesin translocation). This second model for directed cohesin movement is not mutually exclusive with the 'loop translocation' model. Both models make the prediction that, similar to what is observed during loop extrusion, cohesin is a weak translocating motor that can be stalled by very small forces.

## Possible outcomes of cohesin collisions with DNA-bound obstacles

DNA loop extrusion by cohesin and condensin has so far only been observed in vitro and only using naked DNA substrates. In vivo, DNA is densely decorated by histones and other DNA binding proteins related to transcription, DNA replication and other forms of DNA metabolism. If we portray a loop extruding cohesin complex in its slipping state next to a nucleosome (*Figure 8A*, left), it becomes apparent that a nucleosome is too big to pass through the channel formed between Scc2$^{Mis4}$ and the SMC ATPase heads.

A possible path for nucleosome bypass opens up when the Scc2$^{Mis4}$ cohesin loader transiently dissociates. DNA now passes in and out of cohesin through the Scc3-Smc3-kleisin-N chamber (*Figure 8A*, Nucleosome bypass, and *Figure 8—figure supplement 1A*). If Scc3$^{Psc3}$ disengages from the SMC hinge in the slipping state, the clearance available for obstacle bypass further increases (*Figure 8—figure supplement 1B*). In the case of topologically loaded cohesin, the same channel is in principle wide enough to allow nucleosome bypass, albeit denser nucleosome arrays block purely diffusive cohesin sliding (*Stigler et al., 2016*). During loop extrusion, on one hand, Brownian ratchet-driven directional cohesin movement will facilitate nucleosome bypass. On the other hand, only one DNA lies in the Scc3-Smc3-kleisin-N chamber following topological loading, but both in and outward pointing DNAs must be accommodated during loop extrusion. If both DNAs are histone-bound, especially if these histones form part of higher order structures, considerable steric constraints will be encountered that likely slow down or stop loop extrusion.

An alternative outcome of cohesin-nucleosome collisions is therefore that loop extrusion at least temporarily stalls. As the cohesin loader dissociates, DNA is released from the Scc2-head module and free to move within the Scc3-Smc3-kleisin-N chamber. As Scc2$^{Mis4}$ rejoins the complex, there is a new chance for kleisin N-gate passage during gripping state formation, as originally foreseen during topological loading (*Figure 8A*, Kleisin N-gate passage). If this occurs, the DNA loop resolves following ATP hydrolysis, resulting in cohesin topologically embracing one DNA.

There might be yet another possible outcome of cohesin-nucleosome encounters. Given the thrust of a diffusion-mediated collision between cohesin and an obstacle, the closed kleisin N-gate might rupture (*Figure 8A*, Kleisin N-gate rupture). If this happens, the extruded DNA loop will again resolve, resulting in cohesin topologically embracing one DNA. One can imagine that frequent stalling on a nucleosome-dense chromatin template prevents processive loop extrusion. The encountered obstacles could be seen as triggering a 'proofreading' mechanism, prompting recurring attempts at kleisin-N gate passage, eventually resulting in successful topological cohesin loading.

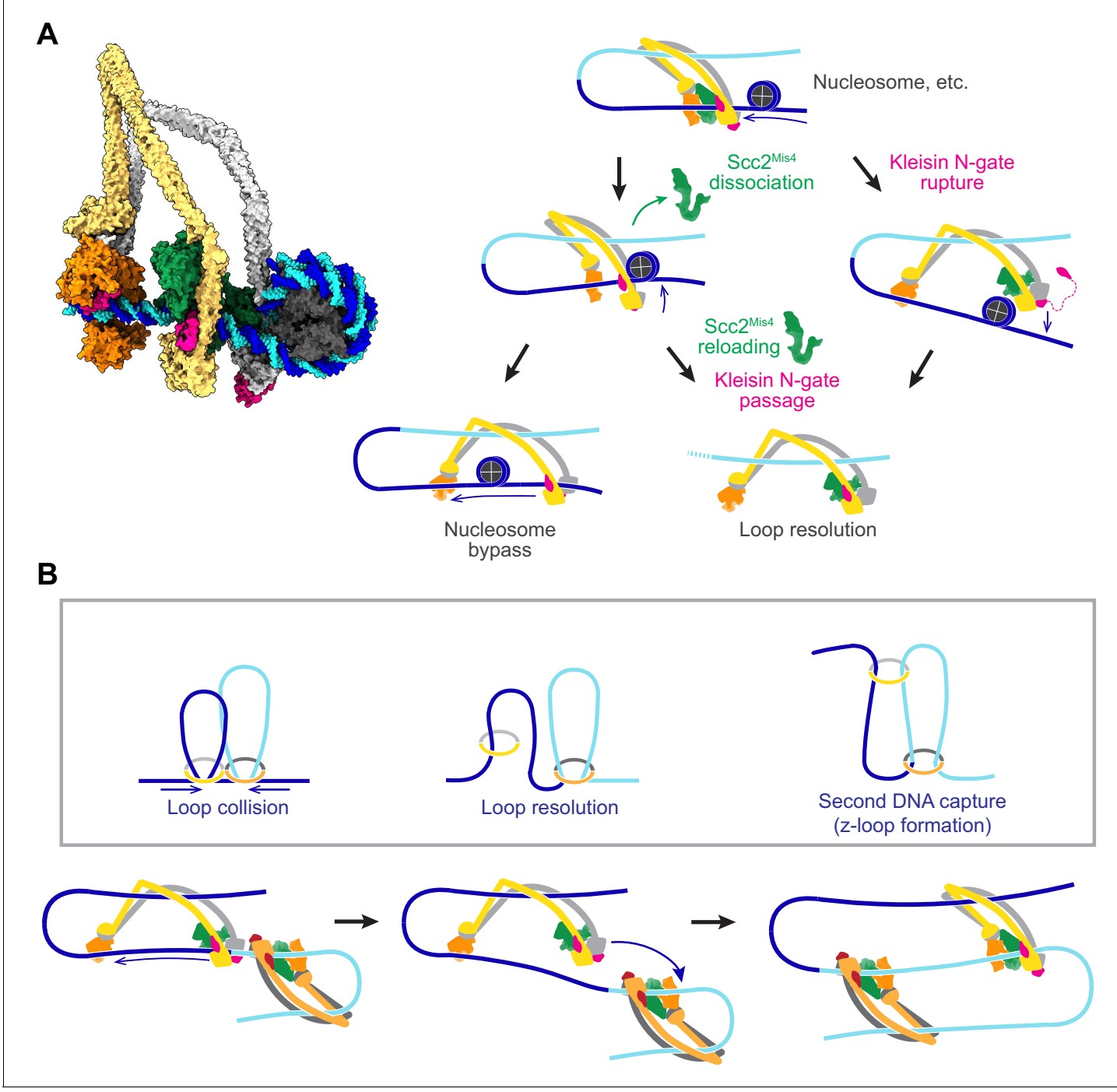

**Figure 8.** Possible outcomes of obstacle encounters. (**A**) Structural model of cohesin in the slipping state encountering a nucleosome (left). Three possible outcomes are shown on the right. If the cohesin loader dissociates, an Scc3-hinge module stroke can achieve *Nucleosome bypass* before the next gripping state forms. Alternatively, cohesin loader dissociation and reloading opens a renewed opportunity for *Kleisin N-gate passage* by DNA, resulting in *Loop resolution*. The impact of a nucleosome collision might alternatively cause *Kleisin N-gate rupture*, which again results in *Loop resolution*. (**B**) A model for z-loop formation following head-on *Loop collision* of two loop extruding condensin complexes. One of the two condensins undergoes *Loop resolution*, by either kleisin N-gate passage or N-gate rupture. The now topologically bound condensin complex diffuses in search for a substrate for *Second DNA capture*, resulting in *z-loop formation*. During two-sided z-loop growth, depicted here, both in and outward pointing DNAs move through both condensin complexes. One is moved by each condensin's Brownian ratchet, the other by the motion of the opposite condensin. The online version of this article includes the following figure supplement(s) for figure 8:

**Figure supplement 1.** Additional views of cohesin during nucleosome bypass.

*Figure 8 continued on next page*

*Figure 8 continued*

**Figure supplement 2.** Schematic that reconciles our molecular model of cohesin function with an alternative model for DNA entry into the cohesin ring, based on chemical crosslinking (*Collier et al., 2020*).

## Z-loop formation by SMC ring complexes

Chromosomal cohesin loading sites lie at considerable distances from each other (*Schmidt et al., 2009*). Nevertheless, we cannot outright dismiss the possibility that loop extruding cohesin complexes might collide in head-on encounters. In vitro observations of colliding, loop-extruding condensins have revealed that these complexes are able to traverse each other to form distinctive z-loop structures (*Kim et al., 2020*). Can this behavior be explained by our molecular model of SMC complex function? Upon encounter, condensins have been observed to pause for a period of time. This is consistent with the behavior of two Brownian ratchets that collide (*Figure 8B*, Loop collision). A way out of this conflict is provided by one of the above-mentioned loop resolution pathways, kleisin N-gate passage or N-gate rupture. Both allow one of the colliding condensins to resolve their loop and turn into a topologically loaded condensin (*Figure 8B*, Loop resolution). The newly gained freedom of movement of the now singly tethered condensin allows it to diffuse. If, akin to how cohesin entraps a second single-stranded DNA (*Murayama et al., 2018*), condensin engages with a second double-stranded DNA that lies beyond the colliding condensin complex, a z-loop is formed (*Figure 8B*, second DNA capture). Various outcomes have been observed during z-loop formation in vitro, including one- and two-sided z-loop growth. It is possible to explain both behaviors if we assume that second DNA capture during z-loop formation results in either topological capture of the second DNA or results in the second DNA entering loop extrusion mode.

## Discussion

Here, we use structures of a nucleotide-bound DNA gripping state (*Higashi et al., 2020*; *Shi et al., 2020*) as the starting point to develop a molecular framework for DNA entry into the cohesin ring, as well as to show how a subtle but important difference in the starting topology turns this gripping state into a Brownian ratchet that can fuel loop extrusion. The resulting model of cohesin function is able to explain numerous experimentally observed features of both topological loading and loop extrusion and makes a number of testable predictions.

## Use of the energy from ATP binding and hydrolysis during topological entry

Any model of cohesin function must explain how the energy from ATP binding and hydrolysis is used to fuel or regulate its activities. ATP binding leads to SMC head engagement and to a conformational change at the Smc3[Psm3] neck that favors kleisin N-gate opening (*Higashi et al., 2020*; *Muir et al., 2020*). However, kleisin N-gate opening might not be hard-wired and it is possible that head engagement occurs sometimes without kleisin N-gate opening (see below). Head engagement creates a composite DNA binding surface on top of the ATPase heads. The next steps in cohesin's reaction cycle are now likely driven by the binding energy of DNA itself. First, DNA establishes contact with the extensive positively charged surface on the ATPase heads. Next, the DNA engages Scc2[Mis4], which turns into its gripping state conformation as its own positive charges embrace the DNA. Together, the DNA and cohesin loader also establish contact with and close the kleisin N-gate.

The DNA-induced Scc2[Mis4] conformational change creates a docking interface for Scc3[Psc3]. The latter subunit joins the complex by concurrently binding the DNA, as well as by recruiting the SMC hinge. The binding energy released from these additional interactions compensates for the energetic cost of introducing a DNA bend, required to reach this configuration. Assembly of the energy-loaded gripping state is now complete; DNA has entered the cohesin ring through the kleisin N-gate. ATP hydrolysis now ensues, which dissolves the gripping state. However, dissolution equates to more than mere dispersal of the components. The geometric arrangement of the Scc3-hinge and Scc2-head modules means that diffusion-driven gripping state dissolution generates a swinging

motion of the Scc3-hinge module that guides DNA through the head gate to complete topological entry into the cohesin ring.

## Use of the energy from ATP binding and hydrolysis during loop extrusion

If DNA did not pass the kleisin N-gate, all other considerations for gripping state assembly remain unchanged. Again, gripping state dissolution initiates Brownian Scc3-hinge swinging motion. However, the kleisin prevents head gate passage, instead resulting in DNA slippage along the Scc2-head module and the initiation of a DNA loop. Based on the observed DNA path, DNA binding to the Scc3-hinge and Scc2-head modules has already introduced a roughly 120° DNA bend in the gripping state. This bend must mature into a 180° turn that fits through the approximately 23 nm clearance of the Scc3-Smc3-kleisin-N chamber. DNA spontaneously adopts narrow bends (*Vafabakhsh and Ha, 2012*) and DNA thermal fluctuations might suffice for loop extrusion to commence, possibly after a certain delay. We cannot exclude that additional mechanisms or DNA-protein contacts contribute to loop initiation, a process that remains to be experimentally examined. Once a DNA loop is formed, its extension is energetically favorable as the loop radius increases. The energetic cost of reducing the loop radius again decreases the chance that a DNA loop slips back, once formed.

A key feature that determines the processivity of the Brownian ratchet is the half-life of the Scc3-hinge module interaction with DNA following the gripping to slipping state transition. This interaction should last long enough to guide directed diffusion but short enough so that DNA is released before the next gripping state forms. Given the stochastic nature of Scc3-hinge module dissociation from, and possible re-association with the DNA, we expect that not all loop extrusion cycles result in productive loop size gain, a fact that might contribute to the wide spread of observed loop extrusion rates. The corresponding component of the Scc3-hinge module in the condensin complex is its putative Ycg1-hinge module. Experiments with condensin harboring a DNA binding site mutation in the Ycg1-hinge module revealed greatly compromised loop extrusion (*Ganji et al., 2018*), consistent with an important role of this element.

Biological motors typically couple ATP-dependent conformational changes to their motion, allowing for robust movements in the presence of counteracting forces (*Howard, 2001*). We cannot exclude the possibility that cohesin also uses an ATP-dependent mechanism to control the relative positions of the Scc3-hinge and Scc2-head modules. For example SMC coiled coil stiffness could store torsional energy during gripping state formation that is released following ATP hydrolysis to add a power stroke to loop extrusion. However, SMC coiled coils appear very flexible (*Eeftens et al., 2016*; *Ryu et al., 2020a*) and as yet there is no evidence for energy coupling between the ATPase heads and hinge. Our computational simulations suggest that a purely diffusion-driven Brownian ratchet recapitulates experimental observations well. In this scenario, the energy from ATP binding and hydrolysis operates a Brownian ratchet, while thermal energy moves the DNA.

Bulk solution measurements suggested that cohesin undergoes on average approximately one ATP hydrolysis cycle per second in the presence of the cohesin loader and DNA. In contrast, our simulations predict that ATP hydrolysis cycles must happen an order of magnitude faster for the Brownian ratchet to reach experimentally observed loop extrusion speeds. These observations are not necessarily incompatible if complex formation between cohesin, the cohesin loader and DNA is a rate-limiting step. In a typical bulk cohesin loading reaction, approximately 20% of DNA is captured by cohesin within an hour of incubation (*Murayama and Uhlmann, 2014*). This illustrates that gripping state formation is likely a slow process in solution. ATP hydrolysis rates might well be substantially greater than average, once a protein-DNA assembly has formed.

## The importance of kleisin N-gate passage

A conserved kleisin N-tail interacts with DNA to ensure kleisin N-gate passage during gripping state formation (*Higashi et al., 2020*). Why then might kleisin N-gate passage sometimes fail, resulting in loop extrusion? A potentially relevant observation is that the biochemical reconstitution of both topological cohesin loading onto DNA (*Murayama and Uhlmann, 2014*), as well as DNA loop extrusion (*Ganji et al., 2018*), are helped by unphysiologically low-salt concentrations. Electrostatic interactions between DNA and cohesin, which characterize the gripping state, are stronger when less salt

competes with them in the incubation buffer. While enhanced DNA-protein contacts likely promote the biochemical reactions, the low ionic strength will also affect protein-protein interactions. For instance, electrostatic interactions contribute to keeping the kleisin N-gate shut, which are augmented in a low salt buffer and could impede N-gate opening. The boosted efficiency of gripping state formation at low-salt concentrations might thus come at the cost of an increased fraction of failed topological entry events.

Even if kleisin N-gate passage is disfavored in a low-salt environment, recurrent cohesin loader dissociation and reassociation events during loop extrusion might eventually permit successful N-gate passage. This outcome would result in topological cohesin loading onto DNA and loop resolution, similar to what we envision might happen when loop extrusion meets an obstacle. How easily the fate of DNA with respect to the kleisin N-gate changes once loop extrusion has started will be important to explore.

## Alternative molecular models of DNA loop extrusion

How does our Brownian ratchet model compare to previously proposed models for DNA loop extrusion? The first proposed, tethered inchworm model (*Nichols and Corces, 2018*), also features the two HEAT repeat subunits as DNA binding elements that perform a scissoring motion while remaining connected by a flexible kleisin linker. Instead of forming head and hinge modules, the HEAT subunits associate with one of the two ATPase heads, each. The HEAT subunit DNA affinity is postulated to change during the ATP hydrolysis cycle such that force from ATP head engagement and resolution is transferred to move DNA. The SMC hinge in turn is tacitly assumed to act as a second DNA anchor, thereby achieving asymmetric DNA loop extrusion. The authors suggest that obstacle bypass is possible in a stepping motion, as the HEAT subunits alternatingly reduce their DNA affinity. In retrospect, the tethered inchworm model shares several components of our Brownian ratchet. A main difference is that its deterministic nature should achieve a defined loop extrusion speed, even against small counteracting forces.

The DNA segment capture model makes use of one major DNA binding site at the ATPase heads (*Marko et al., 2019*), not too dissimilar to the Scc2-head module of our Brownian ratchet. A key difference is that the head module in the segment capture model exerts a power stroke that alters the angle at which the DNA intersects with the SMC complex, thereby initiating DNA looping or promoting loop growth. After the power stroke, DNA affinity at the head module changes following ATP hydrolysis, again not too dissimilar to our Brownian ratchet. Evidence for the proposed power stroke, as well as for the required mechanism by which DNAs change place between hinge and heads between cycles, remains to be sought.

Finally, the scrunching model is similar to our Brownian ratchet in that thermal fluctuations between DNA binding sites at the head and hinge form the basis for loop formation and loop growth (*Ryu et al., 2020a*). However, the DNA trajectories in the two models show opposite polarity. The scrunching model foresees DNA capture by the SMC hinge in the unfolded state and DNA release in the folded state. This requires at least two regulated DNA binding sites, in addition to which the HEAT repeat subunits are thought to form a static DNA anchor. This contrasts with our proposed ratchet where DNA is caught in the folded state and released after unfolding, and where the HEAT subunits are the two dynamic components that control DNA motion.

## Molecular models of topological DNA entry into the cohesin ring

Our model for cohesin function also makes a molecular proposal for how DNA topologically enters the cohesin ring by sequential passage through the kleisin N-gate and then the ATPase head gate in a top-down direction (*Figure 8—figure supplement 2*). DNA arrival from the top is experimentally supported by DNA-protein crosslink mass spectrometry data and by the fact that covalent closure of the ATPase head gate does not block gripping state formation (*Higashi et al., 2020*). Despite this, an alternative model for DNA entry into the cohesin ring was proposed, in which DNA entry starts by bottom-up passage through the ATPase head gate (*Collier et al., 2020*; *Shi et al., 2020*). While this latter model left open the question how topological entry might be completed, an apparent argument for bottom-up DNA passage through the head gate came from experiments to locate the DNA using chemical crosslinkers. Immediately upon cohesin addition to DNA, before topological loading becomes detectable, the DNA takes up a position in which crosslinkers can trap it within

what have been termed engaged-SMC and engaged-kleisin compartments (*Figure 8—figure supplement 2*; *Collier et al., 2020*), a position that can be reached by bottom-up passage through the ATPase head gate. Our model offers an alternative explanation for this positioning. If we imagine DNA approaching from the top, between the SMC coiled coils, it might frequently pass the disengaged ATPase heads, top-down, before the relatively infrequent ATP-dependent series of head engagement and kleisin N-gate opening commences. This approach results in an equivalent DNA topology following crosslinking.

## On the origin of loop extrusion

Could SMC complexes have evolved to be loop extruding Brownian ratchets? If the primordial function of SMC complexes was that of loop extruders, we should expect the loop extrusion mechanism to be conserved in evolutionary ancient SMC complexes. Our model of loop extrusion suggests that cohesin's DNA-interacting HEAT repeat subunits are key components of the Brownian ratchet. These HEAT repeat subunits are relatively modern additions to SMC complexes. Evolutionarily older SMC complexes that are reflected in today's prokaryotic SMC complexes, as well as in the Smc5-Smc6 complex, contain in place of HEAT subunits two smaller *kleisin interacting tandem winged helix elements* (Kite) (*Palecek and Gruber, 2015*). While Kite subunits interact with DNA (*Zabrady et al., 2016*), they show important differences from how HEAT subunits are incorporated into SMC complexes. Kite subunits bind in close proximity to each other to a relatively short kleisin unstructured region (*Woo et al., 2009*; *Bürmann et al., 2013*; *Jo et al., 2021*). This observation makes it hard to imagine that a similar ratchet mechanism, which requires the DNA binding modules to separate from each other, operates in SMC-kite complexes. In vitro DNA loop extrusion by SMC-kite complexes has not yet been observed, while the molecular basis for SMC-dependent chromatin proximities in *B. subtilis*, attributed to loop extrusion (*Wang et al., 2018*), remains to be fully understood. Further biochemical investigations of SMC-Kite complexes (*Kanno et al., 2015*; *Niki and Yano, 2016*) will provide necessary insight into the question whether these complexes act by topologically loading onto DNA or by DNA loop extrusion.

## Implications for in vivo loop extrusion

Cohesin topologically entraps DNA to promote sister chromatid cohesion (*Haering et al., 2008*). It is possible that kleisin N-gate passage is an error-prone event and that accidental failure of kleisin N-gate passage during a topological DNA loading attempt initiates DNA looping. A close-by obstacle would soon stall loop extrusion and allow proofreading in the form of kleisin N-gate passage or N-gate rupture to reinstate topological loading. This scenario portrays loop extrusion as an unwanted, and possibly rare, side effect of topological cohesin loading. Alternatively, was it a cunning evolutionary twist that allowed cohesin to add loop extrusion to its repertoire of DNA acrobatics?

An obvious challenge to DNA loop extrusion in vivo is the presence of histones and other DNA binding proteins. Our model predicts that obstacle bypass by loop extruding SMC complexes is possible, but also that obstacles reduce the speed and processivity of extrusion, especially when present at a high density on both the in- and outward pointing DNAs. Recent studies reported DNA compaction of histone-bound DNA by cohesin and condensin, but whether SMC complexes indeed bypassed histones in these experiments is not yet known (*Kim et al., 2019*; *Kong et al., 2020*). DNA loop extrusion was also observed in *Xenopus* egg extracts, but loop extrusion was detectable only following histone depletion (*Golfier et al., 2020*). Obstacle encounters during in vitro loop extrusion are an obvious and important area for experimental investigation.

Many cellular DNA transactions make use of histone chaperones and chromatin remodellers to navigate the nucleosome landscape. Indeed, in vitro reconstituted chromosome assembly using purified histones and condensin depends on the histone chaperone FACT (*Shintomi et al., 2015*). While FACT loosens histone-DNA interaction, this chaperone does not possess catalytic activity. FACT typically acts together with much slower-moving RNA or DNA polymerases. Whether FACT can facilitate nucleosome eviction by a Brownian ratchet remains to be explored. Other than FACT, ATP-dependent chromatin remodelers could aid in vivo loop extrusion. When studying the contribution of histone chaperones and chromatin remodelers, we have to keep in mind that topological cohesin

and condensin loading onto chromosomes also requires free DNA access that these enzymes provide (*Toselli-Mollereau et al., 2016*; *Garcia-Luis et al., 2019*; *Muñoz et al., 2019*).

DNA loop extrusion by SMC complexes is a captivating molecular event that provides an at first sight intuitive explanation for chromosome loop formation. However, DNA extrusion is not the only explanation for loop formation. Cohesin and condensin could alternatively generate loops simply by sequential topological capture of two DNAs that come into proximity by Brownian motion, a mechanism that we refer to as diffusion capture (*Cheng et al., 2015*; *Gerguri et al., 2021*). When cohesin is depleted and re-supplied to human cells, small and large loops form with similar kinetics (*Rao et al., 2017*), a behavior that is more readily explained by a diffusion-mediated process than by gradual loop growth.

In addition to cohesin and condensin, weak diffusion-driven motors, chromosomes harbor abundant, strong DNA translocases in the form of RNA polymerases. These are known to push SMC complexes as they move along chromosomes during gene transcription (*Lengronne et al., 2004*; *Davidson et al., 2016*; *Ocampo-Hafalla et al., 2016*; *Busslinger et al., 2017*). We have suggested that, following loop formation by diffusion capture, RNA polymerases provide extrinsic motor activity that promotes loop expansion (*Uhlmann, 2016*). Such transcription-dependent extrinsic loop growth could explain chromatin domain features in an analogous fashion to cohesin moving on its own accord (*Bailey et al., 2020*). The role of RNA polymerase-dependent SMC complex movements in chromosome architecture deserves further attention.

## Outlook

Our molecular proposal for SMC complex function informs the evaluation how topological loading onto DNA and loop extrusion by SMC complexes contribute to chromosome function. While topological loading and loop extrusion share many reaction steps, the two mechanisms also differ. The next challenge will be to exploit these differences to engineer SMC complexes that can topologically load onto DNA but not loop extrude, and vice versa. This will eventually enable genetic experiments that clarify the respective physiological contributions of topological loading and loop extrusion by SMC complexes.

## Materials and methods

**Key resources table**

| Reagent type (species) or resource | Designation | Source or reference | Identifiers | Additional information |
|---|---|---|---|---|
| Strain, strain background (*S. cerevisiae*) | *MATa URA3::pGAL1-psm3-3Pk-7his/pGAL10-psm1, LEU2::pGAL1-rad21-HA-pps-protein A/pGAL10-7his-psc3, pep4Δ::HIS3* | *Murayama and Uhlmann, 2014* | Y4443 | Wild type cohesin purification |
| Strain, strain background (*S. pombe*) | *pMis4-pps-protein A (LEU2), pSsl3(ura4+), h-, lue1-32, ura4-D18* | *Murayama and Uhlmann, 2014* | Y4483 | Mis4-Ssl3 purification |
| Strain, strain background (*S. cerevisiae*) | *MATa URA3::pGAL1-psm3-3Pk-7his/pGAL10-psm1(R593-SNAP-P594), LEU2::pGAL1-rad21-HA-pps-protein A/pGAL10-7his-psc3-CLIP, pep4Δ::HIS3* | This study | | Smc1[Psm1]hinge/Scc3[Psc3]-C FRET construct purification |

*Continued on next page*

*Continued*

| Reagent type (species) or resource | Designation | Source or reference | Identifiers | Additional information |
|---|---|---|---|---|
| Strain, strain background (*S. cerevisiae*) | *MATa URA3::pGAL1-psm3-3Pk-7his/pGAL10-psm1,LEU2::pGAL1-rad21-HA-pps-protein A/pGAL10-7his-psc3-CLIP, pep4Δ::HIS3* | This study | | Scc3$^{Psc3}$-C(CLIP) FRET construct purification |
| Strain, strain background (*S. cerevisiae*) | *MATa URA3::pGAL1-psm3-3Pk-7his/pGAL10-psm1,LEU2::pGAL1-rad21-HA-pps-protein A/pGAL10-7his-psc3-SNAP, pep4Δ::HIS3* | This study | | Scc3$^{Psc3}$-C(SNAP) FRET construct purification |
| Strain, strain background (*S. pombe*) | *pCLIP-Mis4 N191-pps-protein A (LEU2), h-, lue1-32, ura4-D18* | This study | | Scc2$^{Mis4(N191)}$-N FRET construct purification |
| Antibody | anti-V5(Pk) (Mouse monoclonal) | Bio-Rad | Cat# MCA1360 | (1:20,000) |
| Antibody | anti-HA (Mouse monoclonal) | Sigma-Aldrich | Cat# 11583816001 | (1:20,000) |
| Antibody | Anti-mouse IgG-HRP (Sheep polyclonal) | GE Healthcare | Cat# NXA931 | (1:20,000) |
| Antibody | Anti-Digoxigenin-AP, Fab fragments (Sheep polyclonal) | Roche | Cat# 11093274910 | (1:30) |
| Recombinant DNA reagent | Lambda DNA | New England BioLabs | Cat# N3013L | |
| Chemical compound, drug | SNAP-Surface Alexa Fluor 647 | New England BioLabs | Cat# S9136S | |
| Chemical compound, drug | CLIP-Surface 547 | New England BioLabs | Cat# S9233S | |
| Chemical compound, drug | SYBR Gold Nucleic Acid Gel Stain | ThermoFisher | Cat# S11494 | |
| Chemical compound, drug | ATP | Sigma-Aldrich | Cat# A2383 | |
| Chemical compound, drug | ADP | Sigma-Aldrich | Cat# A2754 | |
| Chemical compound, drug | Beryllium sulfate tetrahydrate | VWR international LTD | Cat# 16104.14 | |
| Chemical compound, drug | Sodium fluoride 0.5 M Solution | Sigma-Aldrich | Cat# 67414–1 ML-F | |
| Chemical compound, drug | Dynabeads Protein A | ThermoFisher | Cat#10002D | |
| Chemical compound, drug | Protease K | TaKaRa | Cat# 9034 | |
| Chemical compound, drug | Pluronic F127 | Sigma-Aldrich | Cat# P2443 | |
| Chemical compound, drug | β-Casein from bovine milk | Sigma-Aldrich | Cat# C6905 | |
| Chemical compound, drug | DIG-11-dUTP | Jena Bioscience | NU-803-DIGXS | |
| Chemical compound, drug | SYTOX Orange Nucleic Acid Stain | Invitrogen | Cat# S11368 | |

*Continued on next page*

*Continued*

| Reagent type (species) or resource | Designation | Source or reference | Identifiers | Additional information |
| --- | --- | --- | --- | --- |
| Software, algorithm | UCSF ChimeraX | https://www.cgl.ucsf.edu/chimerax/ | | |
| Software, algorithm | PyMOL | https://pymol.org/2/ | | |
| Software, algorithm | SWISS-MODEL | https://swissmodel.expasy.org | | |
| Software, algorithm | CCBuilder 2.0 | http://coiledcoils.chm.bris.ac.uk/ccbuilder2/builder | | |
| Software, algorithm | Fiji Image J | https://imagej.net/Fiji | | |

## Molecular model of the cohesin complex

The molecular model of cohesin in this study is based on our cryo-EM structure of the fission yeast cohesin complex together with its loader in the nucleotide-bound DNA gripping state (PDB: 6YUF) (*Higashi et al., 2020*). A molecular model of the fission yeast SMC hinge domain was obtained based on a mouse cohesin hinge crystal structure as a template (PDB:2WD5) (*Kurze et al., 2011*) using SWISS-MODEL (*Waterhouse et al., 2018*). Scc3$^{Psc3}$ with the kleisin middle region, bound to DNA, was modeled based on a crystal structure of the corresponding budding yeast components (PDB: 6H8Q) (*Li et al., 2018*). The hinge and Scc3$^{Psc3}$ were manually placed so that they align with the respective positions of the SMC hinge and Scc3$^{SA1}$ in the cryo-EM structure of human cohesin in the gripping state (PDB: 6WG3) (*Shi et al., 2020*). The indicative position of Scc3$^{Psc3}$ in the fission yeast structure, shown for comparison, was estimated based on distance constraints from a protein crosslink mass spectrometry dataset and guided by the negative staining EM density (*Higashi et al., 2020*). The coiled coils emanating from the ATPase heads were extended toward the SMC hinge using modeled Smc1$^{Psm1}$ and Smc3$^{Psm3}$ coiled coil segments, built based on their amino acid sequence using CCbuilder2.0 (*Wood and Woolfson, 2018*). The elbow positions in Smc1$^{Psm1}$ and Smc3$^{Psm3}$ were previously identified (*Higashi et al., 2020*).

To build a molecular model of cohesin in the post-hydrolysis slipping state, Scc2$^{Mis4}$ and the Smc3$^{Psm3}$ head domain were replaced with models of the same elements in new conformational forms, corresponding to previously observed free crystal structure states, as described (*Gligoris et al., 2014*; *Kikuchi et al., 2016*; *Higashi et al., 2020*). To model Brownian motion of the Scc3-hinge module relative to the Scc2-head module, Scc3$^{Psc3}$ with the kleisin middle region and DNA together with the SMC hinge were considered to be one rigid body. The position of this Scc3-hinge module was then developed using a swinging motion of the SMC coiled coils with the inflection point at their elbows.

The molecular model of a nucleosome is based on the crystal structure of a human nucleosome (PDB: 3AFA) (*Tachiwana et al., 2010*). All structural figures were prepared using Pymol (Schrödinger) and ChimeraX (*Goddard et al., 2018*).

## Protein purification and labeling

For the construction of cohesin complexes harboring FRET reporters, SNAP- and CLIP-tag sequences were introduced into the YIplac211-Psm1-Psm3 and YIplac128-Rad21-Psc3 budding yeast expression vectors, as well as the pMis4(N191)-PA fission yeast expression vector that were previously described (*Murayama and Uhlmann, 2014*; *Chao et al., 2015*; *Higashi et al., 2020*). For labeling the Psm1 hinge, the SNAP tag sequence was inserted between Psm1 amino acids R593 and P594. Psc3 was fused to the SNAP or CLIP tag sequence at its C-terminus. For labeling Mis4, the CLIP tag sequence preceded the Mis4(N191) N-terminus. All proteins were purified and labeled with BG-surface Alexa 647 and BC-surface Dy547 dyes (New England Biolabs) as previously described (*Murayama and Uhlmann, 2014*; *Higashi et al., 2020*). The absorbance spectra of the labeled

proteins were recorded between 220–800 nm in 1 nm increments using a V-550 Spectrophotometer (Jasco). The concentrations of protein, Dy547 and Alexa 647 were determined from the absorbance at 280, 550, and 650 nm, respectively, and the labeling efficiencies calculated (assuming molar extinction coefficient ($\varepsilon$) for Dy547 and Alexa 647 of 150,000 $M^{-1}$ $cm^{-1}$ and 270,000 $M^{-1}$ $cm^{-1}$, respectively). Topological DNA loading assays to confirm the biochemical activity of the cohesin fluorophore fusions were performed as previously described (*Higashi et al., 2020*).

## Bulk FRET measurements

Bulk FRET measurement were performed as previously described (*Higashi et al., 2020*), with minor modifications. All fluorescence measurements were carried out in reaction buffer (35 mM Tris-HCl pH 7.5, 0.5 mM TCEP, 25 mM NaCl, 1 mM $MgCl_2$, 15% (w/v) glycerol and 0.003% (w/v) Tween 20). Forty µl of reaction mixtures containing 37.5 nM of the respective labeled cohesin, 100 nM Mis4-Ssl3 or 12.5 nM Dy547-labeled Mis4(N191) and 10 nM pBluescript KSII(+) as the DNA substrate were mixed and the reaction was started by addition of 0.5 mM ATP. Alternatively, 0.5 mM ADP and 0.5 mM $BeSO_4$ +10 mM NaF were included to generate the DNA gripping state. The reactions were incubated at 32℃ for 20 min. The samples were then applied to a 384-well plate and fluorescence spectra were recorded at 25℃ on a CLARIOstar Microplate Reader (BMG LABTECH). Samples were excited at 525 nm and emitted light was recorded between 560–700 nm in 0.5 nm increments. To evaluate FRET changes caused by cohesin's conformational changes across different experimental conditions, we report relative FRET efficiency, $I_A/(I_D + I_A)$, where $I_D$ is the donor emission signal intensity at 565 nm resulting from donor excitation at 525 nm and $I_A$ is the acceptor emission signal intensity at 665 nm resulting from donor excitation. To obtain a baseline of apparent FRET due to spectral overlap, we mixed singly Dy547 and Alexa 647-labeled cohesin at concentrations of 12.5 nM and 37.5 nM, respectively, to reflect the fluorophore stoichiometry of the double labeled complex.

## Co-immunoprecipitation

100 nM cohesin labeled at the Psm1 hinge with Alexa 647, 100 nM Mis4ΔN191 labeled at the N-terminus with Dy547, 10 nM pBluescript KSII (+) DNA and 0.5 mM ATP or 0.5 mM ADP/0.5 mM $BeSO_4$/10 mM NaF were incubated in 15 µl of reaction buffer at 32℃ for 20 min, then diluted with 100 µl of reaction buffer. 5 µl of anti-Pk antibody (Bio-Rad, MCA1360)-bound Protein A conjugated magnetic beads were added to the reactions and rotated at 25℃ for 10 min. The beads were washed twice with 500 µl of reaction buffer and the recovered proteins were analyzed by SDS-PAGE followed by in gel fluorescence detection, as well as immunoblotting with the indicated antibodies.

## DNA loop extrusion by the fission yeast cohesin complex

Flow cells were assembled using a piece of parafilm in which a microfluidic channel (1.5 × 40 mm) was pre-cut, sandwiched between a glass slide and a salinized cover slip (Marienfeld, High-precision, 24 × 60 mm). Two holes were drilled in a glass slide and a metal tubing (New England Small Tube Corp) was glued using an epoxy glue (Devcon) to form an inlet and an outlet of the flow cell. Slides were cleaned by sequential 5 min sonication in ethanol, 1 M NaOH and purified water, then baked at 100℃ for 10 min on a heating plate and finally plasma cleaned for 5 min. Cover slips were first cleaned by sequential sonication in acetone, ethanol, 0.1 M NaOH and purified water (5 min for each step), then blow-dried using compressed air and plasma cleaned for 5 min. Subsequently the cover slip surface was silanized by incubating for 1 hr in 5% dichlorodimethylsilane (Sigma-Aldrich) dissolved in heptane. Finally, cover slips were washed by 10 min sonication in chloroform followed by 10 min sonication in purified water and blow-dried using compressed air.

Flow cells were incubated with anti-Digoxigenin antibodies (Roche, 150U) diluted 30 times in PBS for 20 min and washed with 400 µl PBS. To prevent non-specific binding of DNA and proteins, the surface of the flow cell was passivated by incubating with Pluronic F127 (Sigma-Aldrich, 1% solution in PBS) for 10 min followed by PBS wash and incubating with β-Casein (Sigma-Aldrich, 10 mg/ml in PBS) for at least 1 hr. Subsequently the flow cell was washed with PBS and equilibrated with 50 µl of buffer W (0.1 mg/ml β-Casein in PBS). Forty µl of 10 pM of Digoxigenin-labeled λ-phage DNA in buffer W were introduced into the flow cell, incubated for 15 min and washed with 60 µl of buffer W at 4 µl/min using a syringe pump (Harvard Apparatus, Pico Plus Elite 11). The flow cell was further

equilibrated with 150 µl of buffer R (40 mM Tris-HCl pH 7.5, 50 mM NaCl, 2 mM MgCl₂, 5 mM ATP, 10 mM DTT, 250 nM Sytox Orange, 0.2 mg/ml glucose oxidase, 35 µg/ml catalase and 4.5 mg/ml dextrose) at 15 µl/min. Fission yeast cohesin and cohesin loader were introduced at 5 nM and 10 nM concentration, respectively, in buffer R at 15 µl/min. Individual DNA molecules stained with Sytox Orange were imaged at two frames per second using a Nikon Eclipse Ti2 commercial TIRF microscope. Flow cells were illuminated with a 561 nm laser through a Nikon SR HP Apo TIRF 100x/1.49 oil immersion objective. Images were collected using a sCMOS camera (Photometrics, Prime 95B) and saved as TIFF files without compression for further analysis using FIJI ImageJ. All experiments were performed at room temperature.

λ-phage DNA was terminally labelled with Digoxigenin in a 50 µl reaction containing 0.25 mg/ml λ-phage DNA (NEB), DNA Taq polymerase (NEB), dATP, dCTP, dGTP (Promega), dUTP-Digoxigenin (Jena Bioscience) and 1X Standard Taq buffer (NEB). The mixture was incubated at 72°C for 30 min and cleaned up to remove unincorporated nucleotides using a spin-column (Bio-Rad, Micro Bio-Spin P30). The concentration of the final product was assessed by measuring absorbance at 260 nm using a Nanodrop ND1000 spectrophotometer. Digoxigenin-labelled λ-phage DNA was aliquoted and stored at −20°C.

Loop extrusion rates were determined using FIJI ImageJ software as described (*Davidson et al., 2019*). The length of each DNA molecule was manually measured before loop extrusion to convert distance in pixels into kbp. During loop extrusion the length of DNA not contained inside the loop was measured manually, converted into kbp and subtracted from the full length of Lambda DNA (48.5 kbp) to calculate the size of the DNA loop. The loop extrusion rate was determined as the slope of a linear fit to the experimental loop growth data.

## Mathematical modeling of loop extrusion
### DNA model
We based modeling of our DNA on a dssWLC model, as described in *Koslover and Spakowitz, 2014*. The DNA is defined by a sequence of beads with positions $\vec{r}_i$ and an orientation unit vector $\vec{u}_i$ attached to each bead. The energy for a particular chain configuration is given by:

$$E\left(\left\{\vec{r}_i, \vec{u}_i\right\}\right) = \sum_{i=1}^{N} \left[\frac{\epsilon_b}{2}\left|\vec{u}_i - \vec{u}_{i-1} - \eta\vec{R}_i^{\perp}\right|^2 + \frac{\epsilon_{\parallel}}{2}\left(\vec{R}_i \cdot \vec{u}_{i-1} - \gamma\right)^2 + \frac{\epsilon_{\perp}}{2}\left|\vec{R}_i^{\perp}\right|^2\right] \qquad (3)$$

where $\vec{R}_i = \vec{r}_i - \vec{r}_{i-1}$ and $\vec{R}_i^{\perp} = \vec{R}_i - \left(\vec{R}_i \cdot \vec{u}_{i-1}\right)\vec{u}_{i-1}$.

The DNA in this model is split into segments with contour length $\Delta$. The model parameters $\epsilon_b$, $\epsilon_{\parallel}$, $\epsilon_{\perp}$, $\gamma$ and $\eta$ are unambiguous functions of $\Delta$ and polymer persistence length $l_P$ taken from *Koslover and Spakowitz, 2013*. For all our simulations we use $\Delta = 5$ nm and $l_P = 50$ nm.

### Cohesin model
We generated a simplified model of cohesin using five beads representing the Smc1 and Smc3 heads, the Smc1 and Smc3 elbows as well as the hinge. Similar to DNA, each bead is defined by its position and orientation vectors (e.g. $\vec{r}_{Smc3h}$, $\vec{u}_{Smc3h}$ for the Smc3 head, etc). We describe the interaction between hinge, elbow and the corresponding head beads via dssWLCs and the corresponding energy term is given by *Equation (3)*. Based on the experimentally determined elbow position (*Higashi et al., 2020*; *Bürmann et al., 2019*), we choose contour length between the head and elbow to be 30 nm, while the contour length between the elbow and hinge is 20 nm. In order to determine the persistence length of the connecting coiled coil for our simulations, we sampled conformational states of cohesin alone as it transitions multiple times between slipping and gripping states. We found that at the persistence length $l_{CC} = 50$ nm our simulations match the experimentally observed head-to-hinge distance distribution available for the condensin complex (*Ryu et al., 2020b*). A smaller persistence length for condensin's coiled coil of $l_{CC} = 4$ nm was previously reported (*Eeftens et al., 2016*), which included the flexible elbow region that is considered separately in our simulations.

The interaction between the Smc1 and Smc3 heads is treated differently because of their geometry. We describe their interaction with the same dssWLC approach, corrected for their positions and orientations. Interaction energy between the head domains is given by:

$$E_{HH} = \frac{\xi\epsilon_b}{2}\left|\vec{u}_{Smc3h} - \vec{u}_{Smc1h} - \eta\vec{R}_i^{\parallel}\right|^2 + \frac{\xi\epsilon_{\parallel}}{2}\left(\left|\vec{R}_i^{\perp}\right| - \right)^2 + \frac{\xi\epsilon_{\perp}}{2}\left|\vec{R}_i^{\parallel}\right|^2 \tag{4}$$

where $\vec{R}_i = \vec{r}_i - \vec{r}_{i-1}$; $\vec{R}_i^{\parallel} = \left(\vec{R}_i \cdot \vec{u}_{Smc1h}\right)\vec{u}_{Smc1h}$ and $\vec{R}_i^{\perp} = \vec{R}_i - \vec{R}_i^{\parallel}$.

In *Equation (4),* we use the same set of parameters $\epsilon_b$, $\epsilon_{\parallel}$, $\epsilon_{\perp}$, $\gamma$ and $\eta$ as in *Equation (3)* for the SMC coiled coils. However, SMC heads are connected by the cohesin loader subunit, which provides unknown additional stiffness to this connection. To take this into account, we introduce an additional parameter $\xi$. We find that as long as $\xi > 5$, it does not noticeably affect our simulations. Therefore, for most simulations we used a value of $\xi = 10$.

## Interaction between cohesin and DNA

We introduce two separate energy terms describing interactions between DNA and the Smc3 head as well as DNA and the hinge:

$$E_{Smc3-DNA} = \frac{\alpha}{2}\left|\vec{R}_{\perp}\right|^2 + \frac{\beta}{2}\left|\vec{n}\right|^2 + \frac{\gamma}{2}\left|\vec{r}_{Smc3h} - \vec{r}_i\right|^2 \tag{5}$$

where $\vec{R}_{\perp}$ is the shortest distance between Smc3 and the center of the closest DNA bead,

$$\vec{n} = \vec{u}_{Smc3h} \times \left(\vec{r}_{Smc1h} - \vec{r}_{Smc3h}\right)/\left|\vec{r}_{Smc1h} - \vec{r}_{Smc3h}\right| - \vec{u}_i \tag{6}$$

where $i$ is the index of the DNA bead currently interacting with Smc3.

In *Equation (5)*, the first two terms describe the slipping interaction between DNA and Smc3 which allows diffusion of DNA along the Smc3 head. The second term is introduced to take into account the orientation of DNA with respect to the orientation of cohesin. The third term describes a point-to-point gripping interaction. We assume $\gamma = 0$ in the slipping state.

The interaction between the DNA and the hinge is:

$$E_{Hinge-DNA} = \frac{\delta}{2}\left|\vec{r}_{Hinge} - \vec{r}_j\right|^2 \tag{7}$$

where $j$ is the index of the DNA bead currently interacting with the hinge.

*Equation (5)* and *Equation (7)* describe physical bonds between DNA and cohesin. Because these terms grow indefinitely with the relative distance between the two, they do not describe the dynamics of the bond breakage. In order to account for cohesin changes between gripping and slipping states, parameter $\gamma$ was changed to zero for the slipping state. The exact values of parameters $\alpha, \beta, \gamma$ and $\delta$ are inconsequential as they only affect the extent to which the relative distance between cohesin and the DNA fluctuate. We have chosen values of these parameters as a trade-off between minimizing fluctuations to within one spatial discretization step (5 nm) and improving algorithm convergence. A table with all model parameters and their changes between the computational gripping and slipping states can be found in *Supplementary file 1*.

## 3D Monte-Carlo simulations

Numerical calculations were carried out with the Metropolis method for Monte-Carlo simulation (*Heermann, 1990*). Briefly, random beads representing a DNA segment or a part of cohesin is chosen at each step and its position or orientation vectors are randomly modified. Position vectors are modified by adding a random 3D vector with each coordinate distributed evenly on a [-0.75:0.75] nm interval. Orientation vectors are rotated in random direction in 3D by an angle randomly distributed on [-0.8:0.8]. The new full energy of the system $E_{new}$ is calculated. If the new energy is lower than the previous value $E_{new}$, the new state is accepted. If it is larger, the new state Boltzmann factor is calculated: $r = e^{-\frac{E_{new}-E_{old}}{k_B T}}$ and the new state is accepted if $r > p$, where $p$ is drawn from a standard uniform distribution on the interval (0,1).

## Simplified model of diffusion-based loop extrusion

In the simplified model for DNA loop extrusion, a DNA loop inside a cohesin molecule consists of N segments, each 5 nm in length. On each iteration of the algorithm, we consider all possible events that can happen to the system consisting of cohesin and a DNA loop inside. There are five types of events: 1. Random thermal movement of the inbound DNA at the Smc3 head (only possible in the slipping state), 2. Random thermal movement of the outbound DNA, 3. If the hinge is bound to DNA, DNA can move with the hinge if cohesin changes its state from the folded gripping to the unfolded slipping state, or the other way around, 4. The hinge can bind/unbind DNA, 5. Cohesin can change state, from gripping to slipping state, or *vice versa*.

At each iteration, rate constants are calculated for all possible events. The rate constant of DNA diffusion is the rate at which one DNA segment moves one step forward or backward. It is given by:

$$k_{\pm} = \frac{D}{a^2} \tag{8}$$

where $D$ is the diffusion coefficient and $a = 5nm$ length of the DNA segment.

Off-rate of hinge DNA unbinding:

$$k_{Hinge} = \frac{1}{t_{DNA-hinge}} \tag{9}$$

where $t_{DNA-hinge}$ is the lifetime of DNA-hinge interaction defined in the main text.

We assume that the rates of slipping - > gripping and gripping - > slipping state conversion are the same:

$$k_{Cohesin} = \frac{1}{t_{extended}} \tag{10}$$

where $t_{extended}$ is the lifetime of the slipping state defined in the main text.

At a given iteration of the Monte Carlo simulation, we calculate a set of times at which each possible event stochastically occurs:

$$t_i^{(j)} = -\ln(1 - r_i)/k_i \tag{11}$$

where $j$ is the current iteration, $i$ is the given event. $r_i$ is a uniformly distributed number on the interval (0,1), and $k_i$ the rate for event $i$. Then we implement the event which has the smallest $t_i^{(j)}$ and modify the system based on which event occurred. The total time of the simulation is extended by $t_i^{(j)}$.

## Estimation of the diffusion-driven rate of cohesin unfolding

The simplest assumption about the nature of the cohesin transition from the folded gripping to the unfolded slipping state is that it is driven by diffusion only. The diffusion coefficient can be estimated as:

$$D = \frac{k_B T}{6\pi\eta r} \tag{12}$$

where $\eta = 10^{-3}$ Pa·s is the viscosity of water and $r$ is the effective radius of the diffusing protein. The estimate of the time it takes to diffuse distance $x$ is:

$$t = \frac{<x^2>}{2D} \tag{13}$$

The size of the protein is related to its molecular weight. Given cohesin's irregular shape, the diffusion can be better estimated based on the actual size of the protein. Using a size of ~ 20 nm for the Scc3-hinge and Scc2-head modules, we get a conservative estimate of ~ 0.1 ms for the time it will take the modules to separate by diffusion ~ 50 nm.

## Data and software availability

All data generated in this study are included in the manuscript. Materials and resources described in this manuscript will be made available by the authors upon reasonable request. The code used for the molecular-mechanical simulation of behavior and DNA loop extrusion is available in the GitHub repository (https://github.com/FrancisCrickInstitute/CohesinModel, copy archived at swh:1:rev: 62d9088238066810feb6a98e56cc0940fd4943d1, *Higashi, 2021*).

## Acknowledgements

We thank A Costa and S Henrikus for their collaboration on cohesin structural biology and for their advice, B Khatri for discussions, S Kunzelmann and S Mouilleron from the Crick Structural Biology Science Technology Platform for instrument access and advice, and all our laboratory members for discussions and critical reading of the manuscript. Funder Grant reference number Author European Research Council AdG 670412 Frank Uhlmann Boehringer Ingelheim Fonds PhD Fellowship Minzhe Tang The Francis Crick Institute Frank Uhlmann UK Medical Research Council FC001198 Wellcome Trust FC001198 Cancer Research UK FC001198 The Francis Crick Institute Maxim Molodtsov UK Medical Research Council FC001750 Wellcome Trust FC001750 Cancer Research UK FC001750 The funders had no role in study design, data collection and interpretation, or the decision to submit the work for publication.

## Additional information

### Funding

| Funder | Grant reference number | Author |
| --- | --- | --- |
| European Research Council | AdG 670412 | Frank Uhlmann |
| Boehringer Ingelheim Fonds | PhD Fellowship | Minzhe Tang |
| Medical Research Council | FC001198 | Frank Uhlmann |
| Medical Research Council | FC001750 | Maxim Molodtsov |
| Wellcome Trust | FC001198 | Frank Uhlmann |
| Wellcome Trust | FC001750 | Maxim Molodtsov |
| Cancer Research UK | FC001198 | Frank Uhlmann |
| Cancer Research UK | FC001750 | Maxim Molodtsov |

The funders had no role in study design, data collection and interpretation, or the decision to submit the work for publication.

### Author contributions

Torahiko L Higashi, conceived the study, performed the biochemical and FRET experiments and drew the structural figures, wrote the manuscript with input from all coauthors; Georgii Pobegalov, performed the loop extrusion experiments, designed the mathematical model and performed the computational simulations; Minzhe Tang, contributed the concept of second DNA capture by condensin; Maxim I Molodtsov, Frank Uhlmann, conceived the study, wrote the manuscript with input from all coauthors

### Author ORCIDs

Minzhe Tang 🔘 https://orcid.org/0000-0003-2645-8880
Maxim I Molodtsov 🔘 https://orcid.org/0000-0002-3066-0515
Frank Uhlmann 🔘 https://orcid.org/0000-0002-3527-6619

### Decision letter and Author response

Decision letter https://doi.org/10.7554/eLife.67530.sa1
Author response https://doi.org/10.7554/eLife.67530.sa2

## Additional files

### Supplementary files
• Supplementary file 1. Cohesin-DNA interaction parameters in gripping and slipping states. The values and units of the parameters are indicated, together with an indication of changes between the simulated gripping and slipping states.

• Transparent reporting form

### Data availability
The code used for the computational simulations of the cohesin-DNA complex is available in the github repository (https://github.com/FrancisCrickInstitute/CohesinModel (copy archived at https://archive.softwareheritage.org/swh:1:rev:62d9088238066810feb6a98e56cc0940fd4943d1)).

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
