## [Decision Letter]

**Acceptance summary:**

This work combines experiments and simulations together with previously reported biophysical and structural observations to develop a structure-based model that provides mechanistic insight into the two functions of cohesin: cohesion and loop extrusion. This intriguing and informative manuscript will be of broad interest to those working in the fields of chromatin structure, chromosome biology and molecular machines.

**Decision letter after peer review:**

Thank you for submitting your article "A Brownian ratchet model for DNA loop extrusion by the cohesin complex" for consideration by *eLife*. Your article has been reviewed by 4 peer reviewers, one of whom is a member of our Board of Reviewing Editors, and the evaluation has been overseen by Aleksandra Walczak as the Senior Editor. The reviewers have opted to remain anonymous.

Essential revisions:

1) There was a consensus that the manuscript needs to be extensively edited to better introduce the reader to the structures described in the prior work from the group published in 2020. One approach suggested by the reviewers would be to present a series of increasingly complex representations that would help to position the reader to better understand the work, starting with a global structural view of the cohesin complex first that introduces the key elements: hinge, ATPase domains, kleisin and loading factors. From this point representations akin to the current Figure 1—figure supplement 1 will position the reader to fully understand Figure 1. Similarly, introduction of the key terms (e.g. "N gate", the geometry of "entering/exiting the cohesin ring") at the start and ensuring that the helpful general/fission nomenclature is used throughout the text and all Figures would be helpful. Last, clear articulation of what structures arise from high-resolution data versus structural modeling should be clarified.

2) Please clarify what assumptions are made and/or are implicitly defined in the modeling. Specifically, is the sequence of changes between DNA binding and cohesin conformation prescribed and are there any assumptions on whether DNA binding to the hinge domain is dependent (i.e. temporally linked) to the ATPase cycle? How are chemical kinetics combined with the bead dynamic simulation?

3) A common thread in the reviews relates to questions regarding the position, bending, binding and release of the DNA, which need to be addressed. DNA bending is mentioned in the text but it is not discussed quantitatively (and in the context of to the persistence length) particularly with regards to the initiation of loop extrusion. DNA conformation is also not communicated well in the Figures. Further clarification is needed to address what occurs upon DNA release – whether there are concerns about tangling – and discussing what prevents the DNA loop from sliding outwards from the cohesin ring to release the bending strain. Last, the authors should address what rules out the random binding of the DNA to the hinge module in the unfolded cohesin state, which upon locking the hinge and head together could result in a "backward" sliding with DNA being pushed back through the head domain.

4) The authors should discuss how their model impinges on (or is influenced by) the discrepancy between the rates of loop extrusion and measured ATPase activity.

5) The authors should include further discussion of how their work informs on other models for cohesin and loop extrusion. There is value in evaluating these other models more critically based on the findings reported in the manuscript.

6) The authors should address two issues related to the FRET measurements, namely 1) concerns over the sensitivity of the proximity of Scc3-C and Scc2-N and how this impinges on the modeling and 2) the functionality of the SNAP/CLIP-tagged proteins.

*Reviewer #1 (Recommendations for the authors):*

1. The manuscript will benefit from a clearer articulation of what information arises from previously described structures (for example conformation in the "gripping state") and what is structural modeling based on the available separated structural data. These details could be deduced from the description in the Materials and methods but it will help the reader appreciate the authors' arguments if it is stated in the main text.

2. Regarding the mechanism of the loop extrusion, there are some aspects that remain unclear. First, are there any assumptions on whether DNA binding to the hinge domain is dependent (i.e. temporally linked) to the ATPase cycle steps happening at the head domain? This question is relevant, as one might think that absolutely random independent binding of the DNA to the hinge module could also result in DNA binding to the hinge domain in the unfolded cohesin state, in which case locking the hinge and head together may result in a "backward" sliding with DNA being pushed back through the head domain.

3. Related to Point 2, what, if any, assumptions were made about the sequence of changes between DNA binding and cohesin conformation in the Monte-Carlo simulations? In the text, it is stated (ln 384-389) "When we then allow DNA to detach from the Scc3-hinge module and switch cohesin back to the gripping state, the system readily resets and primes itself for the next cycle (Figure 4D). Our simulations reveal that repeated rounds of the states: "gripping → slipping → DNA detachment from the Scc3-hinge module → gripping" results in continuous extrusion of DNA with an average loop size increase of ~ 30 nm per cycle". It is crucial to clarify whether the sequence of events described here was explicitly imposed or whether independent stochastic "gripping-slipping" and "DNA binding/detachment" were also considered in some simulations. In the latter case, what prevents the DNA thread from sliding back? Perhaps a discussion on what sets the directionality of the DNA translocation would help: is it DNA bending, timing of DNA binding at the hinge module, etc?

4. While the authors provide detailed descriptions of the simulations there are some aspects that would benefit from further clarification. For the Monte Carlo simulations, it is not clear how chemical kinetics was combined with the bead dynamic simulation: (1) DNA binding – what determines/switches which D bead is "currently interacting" with hinge or with head beads, (2) ATPase cycle – what determines the timing of switching from gripping to slipping state, (3) do the head beads change their interaction with the hinge bead, and if so what determines the switch? If not, some explanation for how random folding-unfolding results in directed translocation would be very helpful. Regarding the Monte Carlo steps, it is not clear what were allowed as random moves, i.e. from a given state how a new state was generated. For the diffusion-based model, a more detailed description (and perhaps a cartoon of the simulation model) would be helpful. Would it be correct to say that this model has two spatially fixed points through which a DNA polymer can perform a 1D random walk for "diffusion only", with a constant directed polymer translocation through one of the points was added in "diffusion plus ratchet"?

5. The authors suggest that their mechanism generally results in an asymmetric loop extrusion (Figure 6A left). Once DNA unbinds from the hinge module, what prevents the DNA loop from sliding outwards from the cohesin ring to release the bending strain?

6. In the introduction, the authors described three proposed models for loop extrusion; in the discussion it would be helpful to "circle" back and compare the presented model with these prior considerations to highlight commonalities and incompatibilities.

7. The authors describe the role of kleisin position relative to the DNA much later in the text rather than introducing the concept with the slipping state. Explaining it earlier will help in better understanding the terminology and the key difference between the "topological entry" and "loop extrusion" modes.

8. Bars in Figure 2 might be somewhat misleading as their height does not represent values. I would suggest using a scatter plot instead.

9. A few points on terminology: 1) "Loop extrusion by biased Brownian fluctuations" (l.41) – it seems that fluctuations are not biased, but instead repeatedly get caught in one "loaded" conformation, relaxation of which drives the directed translocation. I would consider more precise wording to highlight the beauty of the mechanism. 2) 'kleisin path' (l.318) – perhaps, 'kleisin chain' as path is used to refer to temporal changes in conformations throughout the paper.

*Reviewer #2 (Recommendations for the authors):*

1. In higher eukaryotes, chromatin seems to have clusters of nucleosomes (e.g., PMID: 32967822; PMID: 28712725; PMID: 30044984), but not stretched nucleosome fiber. It might be better to discuss that this nucleosome clustering can be a more severe issue for cohesin's loop extrusion activity.

2. Page 11. "Maturation of the bent DNA in the gripping state.… Brownian motion will have.…" These sentences are unclear to me and should be rephrased.

3. The reference (Suhas et al., 2017) should be Rao et al., 2017 (PMID: 28985562 ).

*Reviewer #3 (Recommendations for the authors):*

1. Please provide a global structural view of the cohesin complex first. It is provided in figure supplement 1 but it has its own problem (see below), and I could not understand Figure 1 until I spent a long time staring at figure supplement 1. For those who are not up to date on cohesin structures (such as me), cohesin is a ring that can open and close at the hinge and at the ATPase domain and DNA passage through the gap in ATPase domain is modulated by kleisin and cohesin loader, and I could not understand figure 1 description and figure 1 because I could not place different parts in the cohesin complex in my imagination.

2. Figure 1 supplement is probably still too complex. Colors are confusing because Rad21 is shown in pink and its color coding is shown above the figure but Psc3 is shown in a similar color and its color coding is missing in the color guide above the figure. Also, fission yeast names are used here instead of the helpful generic^fission^ naming scheme used elsewhere in the paper so it is very challenging to connect figure 1 to figure 1 supplement. The authors should try to reduce mental gymnastics on the part of the readers!

3. Line 272. "The DNA path shown in Figure 3A, panel a, highlights the position of such a bend, based on our DNA-protein crosslink mass spectrometry data (Higashi et al., 2020)."

– Bending of DNA is mentioned but the figures do not make DNA bending obvious. It is not clear where the bend is, looking at the figure. Some quantification is needed to aid the reader in their assessment of this statement, whether there is a significant degree of bending as the authors claim.

4. Line 280. "ATP hydrolysis, resulting in ATPase head gate opening and Scc3-hinge and Scc2-head module uncoupling. This will initiate a swinging motion of the Scc3-hinge module and proximal coiled coil, with a pivot point at the elbow"

– There appears to be a logical jump here. Why would this cause a swinging motion? In addition, a subsequent sentence refers to this movement as Brownian motion, which is even more confusing. Statement such as 'energizing the swinging motion' is not helpful because it is not precise enough.

5. Line 293. 'fast DNA off-rate'. Rate is high or low, not fast or slow, because it is a quantity.

6. I found some of the terminologies confusing. I suspect the readers will have difficult in understanding what an N gate is, for example.

7. Figure 3 description should make it explicit that DNA loop is inserted into the cohesin channel.

8. Figure 3Aa, it appears to me here that DNA is behind both of the yellow protein segments, going up and down, and panel Ab shows a yellow arrow that suggest that something happens so that now one of the yellow segments is not behind the DNA. However, their model, after considering staring at the figures on my part, seems to assume that DNA is simply bent to form a loop and this loop is inserted so that there is no DNA segment behind the yellow protein segments. Figure 3Aa should be redrawn to avoid misleading the readers. Also, what the yellow arrow means needs to be explained.

9. They need to confirm that SNAP/CLIP tags and labeling do not perturb proteins' function. They may have done this in their 2020 Mol Cell paper that introduced the tagged constructs for FRET analysis and if they have done so, it should be mentioned in this manuscript still.

10. Other models for loop extrusion are mentioned in Introduction but it is difficult to tell how they are different among themselves and from the model proposed here just from reading the text. In addition, the authors do not attempt to evaluate other models based on the findings reported here, and that seems to be a missed opportunity for the discussion sect

*Reviewer #4 (Recommendations for the authors):*

The wording "Enter/Exit cohesin ring" is unclear. The authors should explain which side is "enter" and "exit" in the text and/or corresponding figure.

---

## [Author Response]

Essential revisions:1) There was a consensus that the manuscript needs to be extensively edited to better introduce the reader to the structures described in the prior work from the group published in 2020. One approach suggested by the reviewers would be to present a series of increasingly complex representations that would help to position the reader to better understand the work, starting with a global structural view of the cohesin complex first that introduces the key elements: hinge, ATPase domains, kleisin and loading factors. From this point representations akin to the current Figure 1—figure supplement 1 will position the reader to fully understand Figure 1. Similarly, introduction of the key terms (e.g. "N gate", the geometry of "entering/exiting the cohesin ring") at the start and ensuring that the helpful general/fission nomenclature is used throughout the text and all Figures would be helpful. Last, clear articulation of what structures arise from high-resolution data versus structural modeling should be clarified.

We apologise that our introduction was based on too much prior knowledge and that our figures were restricted to only new information. We now provide an introductory structural summary figure (the new Figure 1A). It portrays the different parts of the cohesin complex and how they come together during assembly of the DNA gripping state. The Figure also introduces key terms (kleisin N-gate, hinge, elbow, etc.). The definition of what we mean by DNAs that ‘enter’ or ‘exit’ the cohesin ring during loop extrusion is now also made explicit at the start of their description following from line 491. We have also ensured that generic nomenclature (with fission yeast names in superscript) is used throughout the manuscript and figures. Lastly, we clarify the origin of the different parts of our hybrid cohesin structural model in the rewritten legend to Figure 1—figure supplement 1.

2) Please clarify what assumptions are made and/or are implicitly defined in the modeling. Specifically, is the sequence of changes between DNA binding and cohesin conformation prescribed and are there any assumptions on whether DNA binding to the hinge domain is dependent (i.e. temporally linked) to the ATPase cycle? How are chemical kinetics combined with the bead dynamic simulation?

We thank the reviewers for the opportunity to make our model assumption clearer. We have expanded the description of the model in the methods section as well as revised the main text. Briefly, we used our three-dimensional model to explore how the transition from gripping to slipping state leads to directional DNA loop growth. This model contains no explicit chemical kinetics, and all transitions were prescribed. The system was allowed to equilibrate in each state and then state changes were manually introduced. This is now made clear in the relevant text starting in line 418. We also supply a new Supplementary table 1 that contains all parameters and their values that describe the cohesin-DNA interaction in the gripping and slipping states.

Thus, supplementary video 2 illustrates a “gripping → slipping → DNA detachment from the Scc3-hinge module → gripping” sequence of events that was imposed on our simulations. To simulate this sequence, we initiated the switch from the gripping to slipping state by imposing the change in parameters that describe the equilibrium conformations of these states. Simulations were then continued until the system reached a new equilibrium. After 2 x 10^7^ iterations, sufficient to reach equilibrium, we imposed DNA detachment from the Scc3-hinge module. At this time, we changed the equilibrium parameters back to those of the gripping state and performed simulations further. Once complete, the cycle was repeated. This is clarified in the Supplementary video 2 legend.

In agreement with the prescribed state transition in our simulations, we think it is reasonable to expect that the hinge-DNA interaction is formed as soon as the hinge contacts the ATPase heads in the gripping state and persists for as long as cohesin remains in the gripping state. This assumption is based on the structural observation of aligned Scc2-head and Scc3-hinge modules in the gripping state. Once this ordered state dissolves following ATP hydrolysis (the gripping to slipping state transition), the Scc3-hinge likely continues to interact with DNA for a random amount of time characterized by its in-solution off rate. This behaviour is analysed in Figure 5F, but it differs from our simulations, where we manually impose Scc3-hinge module dissociation from the DNA. These differences are made clear in the rewritten section that discusses Figure 5F in lines 440 and following. We also discuss that, in the unlikely event that the Scc3-hinge module does not detach from the DNA, or reattaches outside the gripping state, the results will be a non-productive loop extrusion cycle. See lines 463, 727 – 737.

3) A common thread in the reviews relates to questions regarding the position, bending, binding and release of the DNA, which need to be addressed. DNA bending is mentioned in the text but it is not discussed quantitatively (and in the context of to the persistence length) particularly with regards to the initiation of loop extrusion. DNA conformation is also not communicated well in the Figures. Further clarification is needed to address what occurs upon DNA release – whether there are concerns about tangling – and discussing what prevents the DNA loop from sliding outwards from the cohesin ring to release the bending strain. Last, the authors should address what rules out the random binding of the DNA to the hinge module in the unfolded cohesin state, which upon locking the hinge and head together could result in a "backward" sliding with DNA being pushed back through the head domain.

The reviewers bring up an important topic for consideration that we have ourselves given much thought. How likely is it that, by mere thermal fluctuation, DNA reaches a 180° bend, small enough to fit through the 23 nm clearance of the Scc3-Smc3-kleisin-N chamber to initiate loop extrusion? We have taken various semi-quantitative approaches, e.g. considering the angle distribution of a polymer with a persistence length *l_P_* = 50 nm and estimating the time it takes to reach the required conformation based on a probable friction coefficient. While many assumptions go into such estimates, they suggest that the required bends do form in relevant timescales. Additionally, empirical observations agree that DNA readily adopts sharp turns (e.g. Vafabakhsh and Ha, Science, 2012). Another relevant consideration is that the highly ordered gripping state presents an entropy minimum and that the entropy gain from its disassembly should at least in part compensate for the energetic cost of loop formation. However, it remains hard to quantitatively translate these estimates to the specific context of the cohesin-DNA complex, incubated in a non-physiologically low ionic strength environment. This is discussed in the revised manuscript, lines 316 – 322 and 712 – 723.

Once a loop is formed, its extension is energetically favorable as the loop radius increases. The energetic cost of reducing the DNA loop radius again decreases the chance that the DNA loop slips back, once formed. This is explained in lines 723 – 726. We do not foresee tangling between DNA and the cohesin ring, as this would require much sharper bending of DNA and protein, compared to the bends discussed above.

To better communicate the issue of DNA bending, we have highlighted the DNA bend in the gripping state in Figure 3, as well as straightening of the bend during topological DNA entry (Figure 3A) or its development into a DNA loop during the initiation of loop extrusion (Figure 3B).

Lastly, we cannot exclude that the Scc3-hinge module sometimes fails to detach from DNA before cohesin returns to its next gripping state. We also cannot exclude that the Scc3-hinge module detaches and then re-attaches, though the latter scenario is less likely, as the correct geometry for reestablishment of the Scc3-hinge module interaction with DNA will be hard to find outside the gripping state. Nevertheless, both possible scenarios suggest that unproductive loop extrusion steps occasionally occur and might contribute to the wide spread of observed loop extrusion rates. This is discussed in lines 727 – 737.

4) The authors should discuss how their model impinges on (or is influenced by) the discrepancy between the rates of loop extrusion and measured ATPase activity.

There is indeed an apparent discrepancy between the low cohesin ATP hydrolysis rates measured in bulk solution experiments and our simulations. The latter predict that ATP hydrolysis cycles must happen an order of magnitude faster for the Brownian ratchet to reach experimentally observed loop extrusion speeds. These observations are not necessarily incompatible. If complex formation between cohesin, the cohesin loader and DNA is a rate limiting step, then the ATP hydrolysis rate of the assembled complex might well be substantially faster than bulk measurements suggest. In a typical bulk topological cohesin loading reaction, approximately 20% of DNA is captured by cohesin within one hour of incubation (Murayama and Uhlmann, 2014). This illustrates that gripping state formation is indeed likely to be a slow process in solution. The ATP hydrolysis rate once a protein-DNA assembly is formed could be substantially higher. This is explained in lines 750 – 760 of the revised manuscript.

5) The authors should include further discussion of how their work informs on other models for cohesin and loop extrusion. There is value in evaluating these other models more critically based on the findings reported in the manuscript.

We are grateful for the opportunity to discuss how our new insight relates to alternative models for topological DNA entry into the cohesin ring and for DNA loop extrusion. There are interesting parallels, but also important differences, between the previous models and our new structure-based model. We contrast and explore these similarities and differences in an added section of our Discussion between lines 785 and 841.

6) The authors should address two issues related to the FRET measurements, namely 1) concerns over the sensitivity of the proximity of Scc3-C and Scc2-N and how this impinges on the modeling and 2) the functionality of the SNAP/CLIP-tagged proteins.

1) It is true that the absolute FRET efficiency reached by the Scc3-C and Scc2-N fluorophore pair is lower than that reached by our other FRET pairs. This is not unexpected, based on our structural modelling, given the predicted larger Euclidean distance between the Scc3-C and Scc2-N fluorophores. Despite the lower absolute values, the relative FRET efficiency of the Scc3-C and Scc2-N fluorophore pair does increase in the DNA gripping state, consistent with the operation of our proposed Brownian ratchet. This is better explained in the revised manuscript, lines 230 – 245.

2) Following the reviewers’ pertinent suggestion, we now show control experiments that confirm the biochemical activity of each of our fluorophore-tagged cohesin and cohesin loader variants. All the tagged and labelled variants retained the ability to promote topological cohesin loading onto DNA in an ATP and cohesin loader-dependent manner, albeit some of them at a slightly reduced efficiency. This is shown in a new Figure 2—figure supplement 1B.

Reviewer #1 (Recommendations for the authors):1. The manuscript will benefit from a clearer articulation of what information arises from previously described structures (for example conformation in the "gripping state") and what is structural modeling based on the available separated structural data. These details could be deduced from the description in the Materials and methods but it will help the reader appreciate the authors' arguments if it is stated in the main text.

Please confer Essential Revisions, point 1.

2. Regarding the mechanism of the loop extrusion, there are some aspects that remain unclear. First, are there any assumptions on whether DNA binding to the hinge domain is dependent (i.e. temporally linked) to the ATPase cycle steps happening at the head domain? This question is relevant, as one might think that absolutely random independent binding of the DNA to the hinge module could also result in DNA binding to the hinge domain in the unfolded cohesin state, in which case locking the hinge and head together may result in a "backward" sliding with DNA being pushed back through the head domain.

Please confer Essential Revisions, point 2.

3. Related to Point 2, what, if any, assumptions were made about the sequence of changes between DNA binding and cohesin conformation in the Monte-Carlo simulations? In the text, it is stated (ln 384-389) "When we then allow DNA to detach from the Scc3-hinge module and switch cohesin back to the gripping state, the system readily resets and primes itself for the next cycle (Figure 4D). Our simulations reveal that repeated rounds of the states: "gripping → slipping → DNA detachment from the Scc3-hinge module → gripping" results in continuous extrusion of DNA with an average loop size increase of ~ 30 nm per cycle". It is crucial to clarify whether the sequence of events described here was explicitly imposed or whether independent stochastic "gripping-slipping" and "DNA binding/detachment" were also considered in some simulations. In the latter case, what prevents the DNA thread from sliding back? Perhaps a discussion on what sets the directionality of the DNA translocation would help: is it DNA bending, timing of DNA binding at the hinge module, etc?

Please confer Essential Revisions, point 2.

4. While the authors provide detailed descriptions of the simulations there are some aspects that would benefit from further clarification. For the Monte Carlo simulations, it is not clear how chemical kinetics was combined with the bead dynamic simulation: (1) DNA binding – what determines/switches which D bead is "currently interacting" with hinge or with head beads, (2) ATPase cycle – what determines the timing of switching from gripping to slipping state, (3) do the head beads change their interaction with the hinge bead, and if so what determines the switch? If not, some explanation for how random folding-unfolding results in directed translocation would be very helpful. Regarding the Monte Carlo steps, it is not clear what were allowed as random moves, i.e. from a given state how a new state was generated. For the diffusion-based model, a more detailed description (and perhaps a cartoon of the simulation model) would be helpful. Would it be correct to say that this model has two spatially fixed points through which a DNA polymer can perform a 1D random walk for "diffusion only", with a constant directed polymer translocation through one of the points was added in "diffusion plus ratchet"?

Please confer Essential Revisions, point 2.

5. The authors suggest that their mechanism generally results in an asymmetric loop extrusion (Figure 6A left). Once DNA unbinds from the hinge module, what prevents the DNA loop from sliding outwards from the cohesin ring to release the bending strain?

Please confer Essential Revisions, point 3.

6. In the introduction, the authors described three proposed models for loop extrusion; in the discussion it would be helpful to "circle" back and compare the presented model with these prior considerations to highlight commonalities and incompatibilities.

Please confer Essential Revisions, point 5.

7. The authors describe the role of kleisin position relative to the DNA much later in the text rather than introducing the concept with the slipping state. Explaining it earlier will help in better understanding the terminology and the key difference between the "topological entry" and "loop extrusion" modes.

Please confer Essential Revisions, point 1.

8. Bars in Figure 2 might be somewhat misleading as their height does not represent values. I would suggest using a scatter plot instead.

A scatter plot is now used instead of bar graph in Figure 2C.

9. A few points on terminology: 1) "Loop extrusion by biased Brownian fluctuations" (l.41) – it seems that fluctuations are not biased, but instead repeatedly get caught in one "loaded" conformation, relaxation of which drives the directed translocation. I would consider more precise wording to highlight the beauty of the mechanism. 2) 'kleisin path' (l.318) – perhaps, 'kleisin chain' as path is used to refer to temporal changes in conformations throughout the paper.

We thank reviewer 1 overall for the numerous insightful and helpful suggestions, including these comments on terminology. ‘Biased fluctuations’ is indeed an oxymoron and incorrect. This was replaced with ‘biased Brownian motion’. We have also applied a consistent terminology to describe the temporal movement of DNA through space as its ‘trajectory’ while the spatial positioning of DNA or of the kleisin unstructured regions is referred to as their ‘path’.

Reviewer #2 (Recommendations for the authors):1. In higher eukaryotes, chromatin seems to have clusters of nucleosomes (e.g., PMID: 32967822; PMID: 28712725; PMID: 30044984), but not stretched nucleosome fiber. It might be better to discuss that this nucleosome clustering can be a more severe issue for cohesin's loop extrusion activity.

The reviewer brings up an excellent point. We now state that higher order nucleosome assemblies can be expected to cause even greater impediment to loop extrusion than histones by themselves, referring to the key references by Nozaki et al. 2017 and Xu et al. 2018 in our introduction, lines 107 – 108.

2. Page 11. "Maturation of the bent DNA in the gripping state.… Brownian motion will have.…" These sentences are unclear to me and should be rephrased.3. The reference (Suhas et al., 2017) should be Rao et al., 2017 (PMID: 28985562).

We thank the reviewer for pointing out these shortcomings, which have been rectified.

Reviewer #3 (Recommendations for the authors):1. Please provide a global structural view of the cohesin complex first. It is provided in figure supplement 1 but it has its own problem (see below), and I could not understand Figure 1 until I spent a long time staring at figure supplement 1. For those who are not up to date on cohesin structures (such as me), cohesin is a ring that can open and close at the hinge and at the ATPase domain and DNA passage through the gap in ATPase domain is modulated by kleisin and cohesin loader, and I could not understand figure 1 description and figure 1 because I could not place different parts in the cohesin complex in my imagination.

Please confer Essential Revisions, point 1.

2. Figure 1 supplement is probably still too complex. Colors are confusing because Rad21 is shown in pink and its color coding is shown above the figure but Psc3 is shown in a similar color and its color coding is missing in the color guide above the figure. Also, fission yeast names are used here instead of the helpful generic^fission naming scheme used elsewhere in the paper so it is very challenging to connect figure 1 to figure 1 supplement. The authors should try to reduce mental gymnastics on the part of the readers!

Please confer Essential Revisions, point 1.

3. Line 272. "The DNA path shown in Figure 3A, panel a, highlights the position of such a bend, based on our DNA-protein crosslink mass spectrometry data (Higashi et al., 2020)."– Bending of DNA is mentioned but the figures do not make DNA bending obvious. It is not clear where the bend is, looking at the figure. Some quantification is needed to aid the reader in their assessment of this statement, whether there is a significant degree of bending as the authors claim.

The DNA bend is now highlighted in Figure 3, as well as its straightening following topological loading or its alternative development into a loop during loop extrusion.

4. Line 280. "ATP hydrolysis, resulting in ATPase head gate opening and Scc3-hinge and Scc2-head module uncoupling. This will initiate a swinging motion of the Scc3-hinge module and proximal coiled coil, with a pivot point at the elbow"– There appears to be a logical jump here. Why would this cause a swinging motion? In addition, a subsequent sentence refers to this movement as Brownian motion, which is even more confusing. Statement such as 'energizing the swinging motion' is not helpful because it is not precise enough.5. Line 293. 'fast DNA off-rate'. Rate is high or low, not fast or slow, because it is a quantity.6. I found some of the terminologies confusing. I suspect the readers will have difficult in understanding what an N gate is, for example.7. Figure 3 description should make it explicit that DNA loop is inserted into the cohesin channel.8. Figure 3Aa, it appears to me here that DNA is behind both of the yellow protein segments, going up and down, and panel Ab shows a yellow arrow that suggest that something happens so that now one of the yellow segments is not behind the DNA. However, their model, after considering staring at the figures on my part, seems to assume that DNA is simply bent to form a loop and this loop is inserted so that there is no DNA segment behind the yellow protein segments. Figure 3Aa should be redrawn to avoid misleading the readers. Also, what the yellow arrow means needs to be explained.

We thank the reviewer for pointing out shortcomings in the text and figures, which have been rectified.

9. They need to confirm that SNAP/CLIP tags and labeling do not perturb proteins' function. They may have done this in their 2020 Mol Cell paper that introduced the tagged constructs for FRET analysis and if they have done so, it should be mentioned in this manuscript still.

Please confer Essential Revisions, point 6.

10. Other models for loop extrusion are mentioned in Introduction but it is difficult to tell how they are different among themselves and from the model proposed here just from reading the text. In addition, the authors do not attempt to evaluate other models based on the findings reported here, and that seems to be a missed opportunity for the Discussion section.

Please confer Essential Revisions, point 5.

Reviewer #4 (Recommendations for the authors):The wording "Enter/Exit cohesin ring" is unclear. The authors should explain which side is "enter" and "exit" in the text and/or corresponding figure.

Please confer Essential Revisions, point 1.